# Changes in Sleep Duration and Sleep Timing in the General Population from before to during the First COVID-19 Lockdown: A Systematic Review and Meta-Analysis

**DOI:** 10.3390/ijerph21050583

**Published:** 2024-05-02

**Authors:** Chiara Ceolin, Federica Limongi, Paola Siviero, Caterina Trevisan, Marianna Noale, Filippo Catalani, Silvia Conti, Elisa Di Rosa, Elena Perdixi, Francesca Remelli, Federica Prinelli, Stefania Maggi

**Affiliations:** 1Geriatric Unit, Department of Medicine (DIMED), University of Padova, 35128 Padova, Italy; chiara.ceolin.1@gmail.com (C.C.); caterina.trevisan.5@studenti.unipd.it (C.T.); filippo.catalani@gmail.com (F.C.); 2Department of Neurobiology, Care Sciences and Society, Aging Research Center, Karolinska Institutet and Stockholm University, 17165 Solna, Sweden; 3Neuroscience Institute, Aging Branch, National Research Council, 35128 Padova, Italy; federica.limongi@in.cnr.it (F.L.); marianna.noale@in.cnr.it (M.N.); stefania.maggi@in.cnr.it (S.M.); 4Department of Medical Sciences, University of Ferrara, 44121 Ferrara, Italy; rmlfnc1@unife.it; 5Institute of Biomedical Technologies, National Research Council, 20054 Segrate, Italy; silvia.conti@itb.cnr.it (S.C.); federica.prinelli@itb.cnr.it (F.P.); 6Neuropsychology Lab, Centre for Cognitive Disorders and Dementia IRCCS Mondino Foundation, 27100 Pavia, Italy; elena.perdixi@humanitas.it; 7Department of General Psychology, University of Padua, 35131 Padova, Italy; elisa.dirosa@unipd.it; 8Department of Neurology, IRCCS Humanitas Research Hospital, 20089 Milan, Italy

**Keywords:** sleep duration, sleep timing, bedtime, wake-up time, napping, general population, COVID-19 lockdown, systematic review

## Abstract

Background: The COVID-19 lockdown had a profound effect on everyday life, including sleep health. This systematic review and meta-analysis evaluated changes in quantitative sleep parameters during the first lockdown compared with pre-lockdown in the general population. Methods: A search in scientific databases was performed to identify eligible observational studies from inception to 8 February 2023. We performed a random effects meta-analysis of those studies reporting (a) means of sleep duration, time in bed (TIB), and sleep timing (bedtime and wake-up time); (b) the percentages of atypical sleep duration before and during the lockdown; (c) the percentages of change in sleep duration and sleep timing. Results: A total of 154 studies were included. A small increase in sleep duration (0.25 standardized mean difference, 95% CI 0.180–0.315) was found, with 55.0% of the individuals reporting changes, predominantly an increase (35.2%). The pooled relative risk for sleeping more than 8/9 h per night was 3.31 (95% IC 2.60–4.21). There was a moderately significant delay in sleep timing and a surge in napping. Conclusion: An increase in sleep duration and napping, and delayed sleep timing were observed. High-quality studies should evaluate whether these parameters have now become chronic or have returned to pre-lockdown values.

## 1. Introduction

The COVID-19 pandemic, caused by the SARS-CoV-2 virus, has brought about profound changes globally, impacting public health, society, economy, and daily life for billions. The rapid spread of cases prompted a global health emergency declaration by the World Health Organization (WHO) on 30 January 2020 [1]. Measures such as hand hygiene, face masks, isolation, quarantine, and lockdowns were implemented in order to limit the diffusion of the virus and to mitigate the burden on health systems, causing a drastic shift in social dynamics [2]. Physical distancing and isolation affected daily habits, including work schedules, exposure to natural light, and reduced physical exercise opportunities [3]. Stress levels rose due to fear of the unknown disease, health concerns, and economic repercussions [3]. Changes in routines disrupted daily rhythms and energy balance, affecting various biological clock regulators [4].

Specifically, considering the vulnerability of the sleep system to cognitive-physiological stress (also known as sleep reactivity), sleep health was significantly compromised during the COVID-19 lockdown [5]. In fact, it is known that ruminating on stressful factors can activate processes that disrupt sleep, and that sleeping difficulty during stressful periods promotes repetitive thinking as the inability to fall asleep creates an unstructured stressful period in bed [5]. Sleep disturbances can present in various forms, including insomnia, disrupted sleep, daytime symptoms such as involuntary drowsiness, difficulty falling or staying asleep, delayed bedtime, abnormal sleep behaviors, and nightmares [5]. According to several systematic reviews and meta-analyses, sleep disturbances were common in different segments of the population during the COVID-19 pandemic [6,7,8,9,10].

Our previous systematic review and meta-analysis found a worsened sleep quality and increased sleep disturbances in the general population during the COVID-19 lockdown compared with pre-lockdown levels [11]. The COVID-19 pandemic and confinement measures led to changes in other relevant sleep parameters, such as delayed bedtime and increased sleep duration in the general population [12,13,14]. This pattern has been confirmed by two systematic reviews and meta-analyses evaluating the impact of the COVID-19 pandemic on sleep health [15,16]. Al-Ajlouni reported a negative impact on sleep with an increase in the prevalence of short or long sleep duration among different populations residing in the Middle East and North Africa (MENA) [15]. Cui’s systematic review and meta-analysis uncovered an increase in sleep duration among healthy adults during the COVID-19 lockdown [16]. However, both works have limitations: the former focused exclusively on the MENA region, limiting its generalizability [15], while the latter included only six studies that evaluated changes in sleep duration [16]. Moreover, there are no systematic reviews that investigated changes in other sleep dimensions, such as napping habits and atypical sleep duration.

The current systematic review and meta-analysis aims to overcome these limitations by providing a more comprehensive understanding of the changes in several quantitative sleep parameters during the first COVID-19 lockdown in the general population. We chose to concentrate on the first lockdown period, as investigations conducted thereafter may have been susceptible to the mitigation of restrictions, potentially resulting in varied effects on people’s sleep patterns. In particular, we evaluated changes in sleep duration, time in bed (TIB), sleep timing (bedtime and wake-up time), and napping habits during versus before the first COVID-19 lockdown.

## 2. Materials and Methods

This systematic review and meta-analysis was conducted in accordance with the Preferred Reporting Items for Systematic Reviews and Meta-Analyses (PRISMA) guidelines [17], and it was registered in PROSPERO, CRD42021256378.

### 2.1. Search Strategy and Selection Criteria

We systematically searched four academic electronic databases (PubMed, Cochrane Library, Ebsco, and Web of Science-WOS), a preprint server (MedRxiv), and a gray literature database (OpenGrey) from inception to 28 May 2021; an updated search through 8 February 2023 was also carried out (PubMed and WOS). The full search strategy and the search terms used for each database are described in Appendix A (Table A1). The reference lists of relevant systematic reviews and articles were manually searched for additional studies. All the references were downloaded into Zotero, and this citation manager software was used for every stage of the selection process, from downloading and removing duplicates to screening the abstract titles and the full-texts. The abstract titles and full-texts were screened independently by three authors (E.P., F.R., and F.L.). Any disagreements were solved by consulting the senior authors (S.M., C.T., F.P., and M.N.).

### 2.2. Inclusion and Exclusion Criteria

Original observational cross-sectional or longitudinal studies that assessed changes in sleep characteristics (using self-reported or objective measures) during the first COVID-19 lockdown (hereafter lockdown) compared with before the lockdown (hereafter before) in the general adult population (adults ≥ 18 years) were considered eligible for inclusion.

Studies were excluded if they: (1) were not observational; (2) were reported in languages other than English, Italian, or Spanish; (3) evaluated changes only in subjects with specific diseases (e.g., obesity, diabetes, neuromuscular disease, cancer, osteoarthritis, and dementia) or in specific groups of individuals (e.g., healthcare workers, professional athletes); (4) collected data outside the timeframe of the first lockdown.

### 2.3. Outcomes

The current work is the continuum of a previous systematic review and meta-analysis on sleep quality and sleep disturbances in the general population from before to during the COVID-19 lockdown [11].

In particular, this study focuses on changes in the following quantitative sleep parameters:Sleep duration, i.e., the amount of time that a person sleeps;Sleep timing, which refers to bedtime, the time the person goes to bed, and wake-up time, the time the person awakes in the morning;The total duration spent in bed, namely TIB, encompassing both the time dedicated to sleep and any additional time spent lying in bed, whether awake or in a state of rest;Napping habits that refer to the sleep time beyond the main sleep period (percentage of participants tacking nap, length, and frequency).

### 2.4. Data Extraction

Data were extracted by four authors (F.R., E.P., F.L., and P.S.) using a pre-designed Excel spreadsheet. For each included study, the following information was recorded: first author’s name, year of publication, country, study design, assessment period, outcome, population, sample size, percentage of women, participants’ age (mean, median, or interval/age range), data collection method, type of recruitment, and type of measures utilized. The corresponding author was contacted whenever any study appeared incomplete or needed clarification on the presented data.

### 2.5. Risk of Bias Assessment

The Newcastle Ottawa Scale (NOS) for longitudinal [18] and cross-sectional studies [19] was used by two independent authors (F.R. and E.P.) to evaluate the risk of bias. The NOS allows one to judge a study based on three aspects: the selection of the study groups, the comparability of the groups, and the ascertainment of the outcome of interest. Cross-sectional studies can achieve a score of 0–10, and longitudinal studies can achieve a score of 0–9, with higher scores corresponding to a lower risk of bias. Studies whose NOS < 5 are classified as having a low quality and a high risk of bias [20]. A third author (M.N.) was involved in resolving any discrepancies.

### 2.6. Data Analysis

A meta-analysis was performed for data on outcomes that were sufficiently homogenous in terms of statistical and methodological characteristics. Additionally, a qualitative synthesis was performed to synthesize the findings of the studies that were not included in the meta-analysis. A random-effects meta-analysis was carried out for the sleep duration, TIB, and sleep timing (bedtime and wake-up time) outcomes, using the DerSimonian and Laird method; the studies were weighted according to the inverse of the standard error using the MedCalc Statistical Software version 20.118 [21]. 

Studies reporting the following data were included in the meta-analysis:Mean sleep duration, TIB, and sleep timing before and during the lockdown;The percentages of change in sleep duration (increased, decreased, no change) or in bedtime, and wake-up time (delayed, earlier, no change) during the lockdown vs. before; the percentage of atypical sleep duration (short sleep duration < 7 h/night and long sleep duration > 8 h/night) before and during the lockdown.

For each data type, the effect was expressed as a standardized mean difference (SMD)—this effect was interpreted using Cohen J. 1988 as 0.2 < small < 0.5, 0.5 ≤ medium < 0.8, large ≥ 0.8 [22], proportions, or relative risks. The between-study heterogeneity was analyzed using the I² statistic, where a value of 0% indicates no observed heterogeneity, and higher values show increasing heterogeneity [23]. The publication bias was assessed using Egger’s test [24], and, in case of possible bias (Egger’s *p* ≤ 0.05), we conducted one-study-removed sensitivity analyses.

### 2.7. Subgroup Analysis

Whenever possible, we stratified the meta-analysis by the risk of bias (NOS < 5 vs. NOS ≥ 5) and by country’s area. Countries were grouped into 7 areas: North America (Canada, USA), South America (Argentina, Brazil, Mexico, Peru), Central Asia (Bangladesh, India, Malaysia, Nepal, Pakistan), East Asia (China, Japan, Singapore), West Asia (Iran, Jordan, Kuwait, Lebanon, Saudi Arabia, United Arab Emirates), Europe (France, Germany, Hungary, Netherlands, Poland, Romania, UK), Mediterranean Europe (Cyprus, Greece, Italy, Portugal, Spain, Catalonia, Turkey). T-test and Chi-squared tests were used to compare subgroups.

## 3. Results

Overall, 6289 records were retrieved via databases and registers and 20 additional studies were identified via citation searches and systematic reviews. After the two-steps screening process, a total of 154 records were included (for the PRISMA flow diagram, please see Figure 1). The characteristics of the included studies are shown in Table 1 [25,26,27,28,29,30,31,32,33,34,35,36,37,38,39,40,41,42,43,44,45,46,47,48,49,50,51,52,53,54,55,56,57,58,59,60,61,62,63,64,65,66,67,68,69,70,71,72,73,74,75,76,77,78,79,80,81,82,83,84,85,86,87,88,89,90,91,92,93,94,95,96,97,98,99,100,101,102,103,104,105,106,107,108,109,110,111,112,113,114,115,116,117,118,119,120,121,122,123,124,125,126,127,128,129,130,131,132,133,134,135,136,137,138,139,140,141,142,143,144,145,146,147,148,149,150,151,152,153,154,155,156,157,158,159,160,161,162,163,164,165,166,167,168,169,170,171,172,173,174,175,176,177,178]. Studies were conducted in Argentina (N = 3), Australia (N = 3), Austria (N = 1), Bangladesh (N = 2), Brazil (N = 4), Canada (N = 2), China (N = 6), Colombia (N = 1), Croatia (N = 2), Cyprus (N = 1), Egypt (N = 1), France (N = 6), Germany (N = 2), Greece (N = 4), Hungary (N = 1), India (N = 8), Iran (N = 2), Italy (N = 14), Japan (N = 2), Jordan (N = 2), Kuwait (N = 1), Lebanon (N = 1), Libya (N = 1), Malaysia (N = 1), Mexico (N = 6), Morocco (N = 1), Nepal (N = 1), Netherlands (N = 1), New Zealand (N = 1), Pakistan (N = 3), Peru (N = 1), Poland (N = 4), Portugal (N = 1), Romania (N = 1), Russia (N = 2), Saudi Arabia (N = 5), Singapore (N = 2), South Africa (N = 1), Spain (N = 15), Turkey (N = 1), UK (N = 9), Ukraine (N = 1), United Arab Emirates (N = 2), USA (N = 13), and 12 were carried out in multiple countries. Due to the lockdown, most studies collected data via online surveys. One hundred and forty-five studies used self-reported instruments [25,26,27,28,29,30,31,32,33,34,35,36,37,38,39,40,41,42,43,44,45,46,47,48,49,50,51,53,54,55,56,57,58,59,60,61,62,63,64,65,66,68,90,91,92,93,94,95,96,97,98,99,100,101,102,104,105,106,107,109,110,111,112,113,114,115,116,117,118,119,120,121,123,124,125,127,129,130,131,132,133,134,135,136,137,138,139,140,141,142,143,144,145,147,148,149,150,151,152,153,154,155,156,157,158,159,160,161,162,163,164,165,166,167,168,169,170,171,172,173,174,175,176,177,178]; seven studies used objective measures [67,69,103,108,122,128,146]; and two studies used both [52,126]. One hundred and thirty studies were cross-sectional, whereas twenty-four were longitudinal.

The mean NOS score for the cross-sectional studies was 4.4 (SD = 1.3; range 2−8), and it was 6.2 (SD = 1.3; range 3−9) for the longitudinal ones (Appendix A Table A2). Overall, 53.9% had a good quality and had a low risk of bias (NOS ≥ 5), representing 96% of the longitudinal and 46% of the cross-sectional studies.

### 3.1. Sleep Duration

Among the 132 studies that examined sleep duration, 107 were included in the meta-analysis, and 25 were narratively described [25,48,55,75,80,87,88,102,104,106,108,113,123,126,128,132,141,144,145,156,159,162,163,169,171].

#### 3.1.1. Meta-Analytic Changes in Sleep Duration: Means before and during the Lockdown

The changes in sleep duration were evaluated considering 69 outcomes reported in 44 studies [27,32,39,40,44,45,46,49,50,51,52,57,63,66,68,70,74,81,85,93,95,97,103,107,111,112,114,115,119,120,122,129,130,136,137,143,146,153,160,161,173,176,177,178]. 

The participants reported a small increase in sleep duration (hours) (SMD = 0.25; 95% CI 0.18–0.32; I^2^ = 97.2%) (Figure 2); the analysis did not show a significant publication bias (Egger’s *p* = 0.63).

Subgroups analysis by the risk of bias did not find significant differences. The studies with a low risk of bias produced a higher effect on sleep duration than the overall set of studies, while those with a high risk of bias produced a lower effect. Specifically, the 54 outcomes of the 30 studies [27,32,40,44,49,50,51,52,57,68,70,81,85,93,97,103,107,115,120,122,129,130,136,137,146,153,160,173,176,177] with a low risk of bias (NOS ≥ 5) showed a significant increase of 0.26 SMD in sleep duration (95% CI 0.18–0.34; I^2^ = 97.5%; not significant Egger’s publication bias). Instead, the 15 outcomes of the 14 studies [39,45,46,63,66,74,95,111,112,114,119,143,161,178] with a high risk of bias (NOS < 5) showed a significant increase of 0.22 SMD in sleep duration (95% CI 0.11–0.33; I^2^ = 95.4%; not significant Egger’s publication bias).

#### 3.1.2. Meta-Analytic Changes: Percentage of Change in Sleep Duration

The percentage change in sleep duration during lockdown with respect to before was evaluated by 51 studies [28,29,31,33,38,41,44,46,47,53,54,60,65,71,73,77,78,79,83,85,89,90,91,94,96,97,98,99,101,105,109,121,124,131,133,135,138,139,140,142,150,154,155,157,164,166,168,172,174,175,178]. 

As shown in Figure 3, the random effects model showed that 55.0% (95% CI 49.84–60.07; I^2^ = 99.6%) of the participants reported a change in sleep duration (hours). In particular, 19.7% reported a decrease (95% CI 16.80–22.87; I^2^ = 99.3%) and 35.2% reported an increase (95% CI 32.09–38.44; I^2^ = 99.2). Significant Eggers’s publication bias emerged for the above outcomes, but the sensitivity analyses confirmed the main findings.

Subgroup analysis by risk of bias uncovered significant differences. The studies with a low risk of bias produced a lower percentage of changes in sleep duration compared with the overall studies, while those with a high risk of bias produced a higher percentage. Specifically, the 24 studies [28,31,38,44,60,65,71,73,77,78,83,85,90,91,94,97,99,121,135,157,166,168,174,175] with a low risk of bias (NOS ≥ 5) showed that 48.8% of the participants reported a change in sleep duration (95% CI 41.54–56.12, I^2^ = 99.8%; significant Egger’s publication bias), 30.7% reported an increase (95% CI 26.13–35.41; I^2^ = 99.5%; not significant Egger’s publication bias), and 19.7% reported a decrease (95% CI 15.29–24.47; I^2^ = 99.6%; not significant Egger’s publication bias). Instead, the 27 studies [29,33,41,46,47,53,54,79,89,96,98,101,105,109,124,131,133,138,139,140,142,150,154,155,164,172,178] with a high risk of bias (NOS < 5), showed that 60.8% of the participants reported a change in sleep duration (95% CI 56.64–64.85, I^2^ = 97.7%; not significant Egger’s publication bias), 39.1% reported an increase (95% CI 35.54–42.73; I^2^ = 98.0%; significant Egger’s publication bias), and 19.8% reported a decrease (95% CI 16.78–23.01; I^2^ = 97.3%; not significant Egger’s publication bias). 

Subgroup analysis by country’s area was only possible for the percentages of change in sleep duration. As we can see from Table 2, the percentage of change in sleep duration was very high in South America (72%), lower in East Asia (45%), and it was around values above 50% in other areas. The change concerned the increase in sleep hours in all areas, especially in Central Asia (44%) and South America (40%), and it was around values above 30% in other areas. Significant differences were observed among all areas except North America (30.5%) vs. Europe (30.8%), and East Asia (33.3%) vs. Mediterranean Europe (33.8%).

#### 3.1.3. Meta-Analytic Changes: Percentage of Atypical Sleep Duration before and during the Lockdown

Changes in the percentage of participants with *atypical sleep duration* were evaluated in 18 studies [26,30,34,44,45,59,60,61,62,72,86,92,93,134,138,151,158,167]. 

Compared with the pre-lockdown period, the percentage of participants with atypical sleep duration increased by approximately 17% during the lockdown (pooled relative risk = 1.17, 95% CI 1.04–1.31; I^2^ = 98.2; not significant publication bias) (see Figure 4).

Data analysis showed that the percentage of those participants who slept more than 8/9 h increased during lockdown; a pooled relative risk of 3.31 was observed (95% CI 2.60–4.21; I^2^ = 95.2, not significant publication bias) [26,30,34,44,45,59,60,61,62,72,86,92,93,134,138,151,158].

Data analysis also uncovered that, with respect to before, the percentage of participants who slept less than 6/7 h decreased during lockdown; a pooled relative risk of 0.82 was observed (95% CI 0.73–0.91; I^2^ = 98.7; not significant publication bias) [26,30,32,34,35,44,45,59,60,61,62,72,86,92,93,107,110,112,114,117,134,138,149,158].

### 3.2. Time in Bed

Among the 15 studies that examined TIB, 12 were included in the meta-analysis and 3 were narratively described [69,88,126].

#### Meta-Analytic Changes in Time in Bed: Means before and during the Lockdown

The changes in TIB were evaluated considering 20 outcomes derived from 12 studies [39,56,57,64,70,74,115,122,129,136,160,161]. The participants reported a small increase in TIB (SMD = 0.24; 95% CI 0.151–0.32; I^2^ = 90.6%; not significant Egger’s publication bias) (see Figure 5).

Subgroup analysis by risk of bias showed no significant differences in TIB: the studies with a low risk of bias produced a lower effect compared with the overall studies, while those with a high risk of bias produced a higher effect. Specifically, the 17 outcomes of the 9 studies [56,57,64,70,115,122,129,136,160] with a low risk of bias (NOS ≥ 5) showed a small but significant increase in TIB (SMD 0.26; 95% CI 0.16–036; I^2^ = 91.8%; not significant Egger’s publication bias); while the three studies [39,74,161] with a high risk of bias (NOS < 5) showed a small but not significant lower increase (SMD 0.15; 95% CI −0.03–0.34; I^2^ = 87.1%; not significant Egger’s publication bias).

### 3.3. Sleep Timing

Out of the 45 studies that examined bedtime, 30 were included in the meta-analysis and 15 were narratively described [27,37,42,50,76,88,100,119,126,127,146,156,159,169,171]. Out of the 51 studies that examined wake-up time, 36 were included in the meta-analysis, 14 were narratively described [27,37,42,74,76,88,100,106,119,146,148,156,169,171], and one study provided data both for the meta-analysis and the qualitative synthesis [126].

#### 3.3.1. Meta-Analytic Changes in Bedtime: Means before and during the Lockdown

Changes in bedtime were evaluated considering 45 outcomes reported in 27 studies [36,39,43,49,56,57,63,64,67,68,70,82,85,103,107,115,120,122,125,130,136,137,143,153,160,170,173]. The participants reported a medium significant delay in bedtime (hours) of 0.51 SMD (95%CI 0.38–0.64; I^2^ = 98.5%; not significant Egger’s publication bias) (see Figure 6).

Subgroup analysis by risk of bias uncovered no significant differences. Specifically, the 32 outcomes of the 23 studies [36,43,49,56,57,64,67,68,70,82,85,103,107,115,120,122,125,130,136,137,153,160,173] with a low risk of bias (NOS ≥ 5) showed a small but significant delay of 0.47 SMD in bedtime (95% CI 0.33–0.62; I^2^ = 98.6%; not significant Egger’s publication bias). The four studies [39,63,143,170] with a high risk of bias (NOS < 5) showed that there was a large significant delay of 0.82 SMD in bedtime (95% CI 0.42–1.21; I^2^ = 94.4%; not significant Egger’s publication bias).

#### 3.3.2. Meta-Analytic Changes in Bedtime: Percentage of Change 

The percentage of participants who changed or maintained the same bedtime during the lockdown compared with the pre-lockdown period were evaluated by 14 studies [28,36,46,58,84,85,91,116,121,138,144,157,165,172]. As shown in Figure 7, the random effects model showed that 57.6% (95% CI 44.41–70.27, I^2^ = 99.97; not significant Egger’s publication bias) of the participants reported a change in bedtime. In particular, 42.9% reported a delayed bedtime (95% CI 30.93–55.29; I^2^ = 99.9; not significant Egger’s publication bias) and 11.9% said they went to bed earlier (95% CI 8.79–15.50; I^2^ = 99.1; not significant Egger’s publication bias).

Subgroup analysis by risk of bias revealed significant differences. Specifically, the seven studies [28,36,84,85,91,121,157] with a low risk of bias (NOS ≥ 5) showed that 54.9% of the participants reported a change in bedtime (95% CI 33.32–75.56, I^2^ = 99.9%; not significant Egger’s publication bias), 36.1% said they had a delayed bedtime (95% CI 16.02–59.14; I^2^ = 99.9%; not significant Egger’s publication bias), whereas 10.7% reported an earlier bedtime (95% CI 8.26–13.45; I^2^ = 96.8%; not significant Egger’s publication bias). Instead, the seven studies [46,58,116,138,144,165,172] with a high risk of bias (NOS < 5), revealed that 61.4% of the participants reported a change in bedtime (95% CI 56.16–66.47, I^2^ = 97.8%; significant Egger’s publication bias), 48.8% a delayed bedtime (95% CI 40.51–57.14; I^2^ = 99.5%; not significant Egger’s publication bias), and 13.3% an earlier one (95% CI 6.62–21.81; I^2^ = 99.5%; not significant Egger’s publication bias).

#### 3.3.3. Meta-Analytic Changes in Wake-Up Time: Percentage of Change 

Changes in the wake-up time were evaluated considering 37 outcomes reported in 28 studies [36,39,43,49,50,56,57,63,64,67,68,70,82,85,103,107,115,120,122,125,130,136,137,143,153,160,170,173]. Participants reported a medium significant delay in wake-up time of 0.78 SMD (95% CI 0.64–0.92; I^2^ = 98.5%; not significant Egger’s publication bias) (see Figure 8). 

Subgroup analysis by risk of bias uncovered no significant differences. However, the 33 outcomes of the 21 studies [36,43,49,64,67,68,70,82,85,103,107,115,120,122,125,130,136,137,153,160,173] with a low risk of bias (NOS ≥ 5) showed a medium significant delay in wake-up time of 0.77 SMD (95% CI 0.63–0.92; I^2^ = 98.7%; not significant Egger’s publication bias). The four studies [39,63,143,170] with a high risk of bias (NOS < 5) showed a large significant delay in wake-up time of 0.87 SMD (95% CI 0.33–1.42; I^2^ = 97.0%; not significant Egger’s publication bias).

#### 3.3.4. Meta-Analytic Changes in Wake-Up Time: Percentage of Change 

The percentage change in wake-up time during the lockdown was evaluated by 11 studies [36,46,58,84,85,91,121,126,138,144,172]. As shown in Figure 9, the random effects model showed that 59.3% of the participants reported a change in wake-up time (95% CI 40.26–77.07, I^2^ = 99.9%; not significant Egger’s publication bias). In particular, 45.0% reported a delayed wake-up time (95% CI 28.38–62.30; I^2^ = 99.9%; not significant Egger’s publication bias) and 11.01% reported an earlier one (95% CI 8.35–13.98; I^2^ = 98.5%; not significant Egger’s publication bias).

Subgroup analysis by risk of bias showed significant differences. The studies with a low risk of bias produced lower percentages of change with respect to the overall studies, but those with a high risk of bias produced higher percentages of change. Specifically, the six studies [36,84,85,91,121,126] with a low risk of bias (NOS ≥ 5) showed that 51.3% of the participants reported a change in wake-up time (95% CI 21.75–80.35, I^2^ = 99.9%; not significant Egger’s publication bias), 30.4% reported a delayed wake-up time (95% CI 9.16–57.43; I^2^ = 99.9%; not significant Egger’s publication bias) and 12.0% reported an earlier one (95% 7.26–17.65; I^2^ = 98.9%; not significant Egger’s publication bias). Instead, the five studies [46,58,138,144,172] with a high risk of bias (NOS < 5) revealed that 69.1% of the participants reported a change in wake-up time (95% CI 62.82–74.96, I^2^ = 98.2%; not significant Egger’s publication bias) with 60.1% reporting a delayed wake-up time (95% CI 53.82–66.11; I^2^ = 97.9%; not significant Egger’s publication bias) and 10.2% an earlier one (95% CI 8.07–12.62; I^2^ = 94.2%; not significant Egger’s publication bias). 

### 3.4. Qualitative Synthesis

#### 3.4.1. Synthesis of Sleep Duration 

Twenty-five studies not included in the meta-analysis evaluated changes in sleep duration (see Table 3). Fifteen reported an increase [25,48,75,102,106,108,126,128,141,145,156,159,162,163], four reported no changes [55,88,132,169], and three reported a decrease in sleep duration [80,87,123]. One study found an increase in sleep duration among students but a decrease among office workers [113]; another study reported an increased duration in the young and middle-aged participants but not in the older ones [104], and one study found longer sleep duration in the evening-type chronotype compared to the neither-type and morning-type groups [144]. These findings are consistent with the overall results of our meta-analysis.

#### 3.4.2. Synthesis of Time in Bed

The three studies not included in the meta-analysis regarding changes in TIB found an increase during the lockdown compared to the pre-lockdown period [69,88,126] (see Table 3).

#### 3.4.3. Synthesis of Sleep Timing

Fifteen studies not included in the meta-analysis evaluated changes in the bedtime (see Table 3). Ten reported delayed bedtime [27,37,42,50,76,88,119,159,169,171] and two earlier one [146,156]; two of the studies were unable to detect a clear change [100,127]. Finally, one study found no change in the overall sample but a greater delay in the evening chronotype group compared to the morning one [126].

Sixteen studies not included in the meta-analysis evaluated changes in the wake-up time (see Table 3). Eleven reported a delayed wake-up time [27,37,42,76,88,119,146,148,156,169,171] and one an earlier wake-up time [74]. Two of the studies were unable to identify a clear change [100,147] and two reported no change [106,126].

These findings are in line with the results of the meta-analysis, which found a delayed bedtime and wake-up time.

#### 3.4.4. Synthesis of Napping Habits

The napping habits, evaluated in twenty-three studies, were only narratively described (see Table 3) [33,36,42,63,64,81,84,86,91,100,107,113,118,119,120,129,136,144,145,147,152,165,169]. 

Most studies found an increase in the frequency and length of naps as well as in the percentage of participants taking naps. According to Franceschini, while the good sleepers did not change or reduce the length of naps, the poor sleepers increased it [84]. Finally, Salfi found that a significantly higher proportion of the evening-type chronotype subjects reported changes in their napping habits with respect to the morning-type and neither-type chronotype subjects [144].

## 4. Discussion

This systematic review and meta-analysis uncovered significant changes in several quantitative sleep parameters of the general population from before to during the first COVID-19 lockdown. An increase in both sleep duration and the percentage of individuals with atypical sleep duration was detected. Moreover, the frequency of individuals reporting long sleep duration increased (≥8/9 h), as well as the time in bed and napping habits. Subgroup analysis by country’s area showed an increased sleep duration in all areas considered (North America, South America, Central Asia, East Asia, West Asia, Europe, Mediterranean Europe), prevalently in Central Asia and South America. In addition, we detected significant differences among all areas except for North America vs. Europe and East Asia vs. Mediterranean Europe. The most striking change that emerged from this study regards the sleep–wake cycle. Indeed, more than 40% of the participants reported a significant delay in bedtime and wake-up time.

When reading these data, it is necessary to consider that most of the studies are cross-sectional and have a high risk of bias that can lead to unreliable interpretations. In addition, due to the lockdown restrictions, most of the studies used self-reported data, which tend to be subject to recall and social desirability biases. This issue may have led to a selection bias by excluding non-regular Internet users, such as older people.

The shift to remote work and virtual classes due to the lockdown forced the majority to go slower and extend their timelines, allowing more time for sleep [179,180]. Considering the worldwide prevalence of insufficient sleep, this increased time to sleep could be seen as a beneficial effect of restrictive measures [181]. However, it is unclear whether the increase in sleep duration corresponded to better sleep quality, as some studies reported a worse sleep quality during the lockdown compared with the pre-lockdown levels [11,56]. Moreover, the finding of an increased time in bed should be interpreted with caution. In fact, several works reported a decrease in sleep efficiency, leading to the hypothesis that an increase in the time spent in bed did not necessarily correspond to a longer sleep duration [11,57]. 

Our findings align with previous systematic reviews and meta-analyses [15,16], as well as certain original studies [12,13,14], which have documented extended sleep duration and delayed bedtime patterns within the general population. However, comparing our results across geographical regions poses challenges [13]. For instance, Robbins et al. observed significant variations in sleep duration both before and during the COVID-19 pandemic, noting a lesser increase in Seoul compared to New York City, Los Angeles, London, and Stockholm [13]. When juxtaposed with the evolution of sleep parameters across the different phases of the COVID-19 pandemic, our study reveals several differences. For instance, a longitudinal investigation unveiled a shift in bedtime during the initial lockdown compared to the pre-lockdown period, followed by a reversal during subsequent lockdowns, eventually reverting to pre-pandemic norms. Similarly, the duration of time spent in bed mirrored this trend. Intriguingly, afternoon napping habits remained largely unaffected during these periods of lockdown [64]. These trends suggest an adaptation to the progression of restrictive measures, with researchers hypothesizing that this adaptation stems from the normalization of lifestyle routines. In contrast, delays in both bedtime and wake-up times were noted during both lockdowns compared to the pre-lockdown scenario in the Belgian population, especially among individuals under 24 years old [182]. Another longitudinal study conducted in Italy highlighted a reduction in sleep duration during both the first and second waves of the pandemic [183].

Below, we analyze the potential primary effects on overall health and seek to explore the possible causes of alterations in quantitative sleep parameters.

### 4.1. The Consequences of Quantitative Sleep Parameters Alterations on General Health

Sleep represents an essential biological process for life and optimal health and it is well known that healthy sleep patterns are characterized by adequate duration, good quality, appropriate timing, and regularity, and the absence of sleep disturbances and problems [184]. The changes in quantitative sleep parameters that emerged from this work are of pivotal relevance, given their potentially harmful impact on the immune system, and other health outcomes. Sleep plays a critical role in maintaining the balance of the immune system, and any alterations in its duration and quality can impact its susceptibility to infections and the effectiveness of vaccinations [185,186]. In the context of SARS-CoV-2 vaccination, studies have indicated a positive association between regular sleep duration and antibody levels [185,187]. Specifically, ensuring sufficient sleep, especially within the first week after booster vaccination, is essential for optimal antibody production [188]. However, it is important to consider that while a higher frequency of daytime napping per week initially showed a negative correlation with antibody levels in one study, this association vanished after adjusting for confounding factors, underscoring the intricate nature of the relationship between sleep and post-vaccination immune response [184,187]. 

Alterations in sleep timing may have important consequences on multiple health variables. Later sleep timing in adults has been associated with depression, adverse brain health outcomes, cognitive impairment, obesity, higher cardiometabolic risk, osteopenia and osteoporosis [189]. An earlier bedtime was associated with a higher risk of diabetes, stroke, obesity, hypertension, and cardiovascular diseases (CVDs) [190]. 

A sleep duration of 7–8 h has a beneficial impact on general health with both shorter and longer sleep durations than 7–8 h are associated with poor outcomes such as mortality, diabetes, CVD, coronary heart disease, obesity, and stroke [191,192]. A prolonged time spent in bed appears to be linked to a greater decline in physical function and a higher risk of motor disability in the older population [193]. 

The impact of napping on health is still a debated topic. Indeed, napping seems to have both positive and negative effects on health and its effects appear to depend not only on the duration of the nap itself but also on other factors such as sex and age. Overall, naps showed significant beneficial effects on several cognitive functions [194], but in older people, only short or moderate duration of naps compared with both non-napping and long napping are associated with cognitive benefits in older adults [195]. Depression seems to be associated with long afternoon naps (≥90 min) in middle-aged women, and with short naps (<30 min) in older men [196]. Daytime napping restricted to 30 min/d for adults (aged < 60 years) has no negative effect on cardiovascular health, while it seems not to be beneficial for older adults [197].

### 4.2. Exploring the Causes of Quantitative Sleep Parameters Alterations

Several variables may have contributed to the sleep changes uncovered by this work. One of the most striking involves the modification to the circadian rhythm. These biological rhythms, operating in 24 h cycles, are strongly influenced by external factors such as social patterns, daily routines, and exposure to sunlight [198]. These stimulating signals were markedly altered during the COVID-19 pandemic due to home confinement and profound lifestyle modification, such as remote work and distance learning, which both contributed to bedtime postponement [12,107,173,199]. The rise in stress and anxiety levels during the lockdown, fueled by psychological distress associated with social isolation, financial challenges, and unemployment, along with growing concerns for personal and loved ones’ well-being, may also have had an impact on sleep health [200]. Finally, although digital platforms such as social media helped to keep people connected during the pandemic and lockdown, the sharp increase in social media usage during the pandemic may also have had an important impact on sleep health [201]. Indeed, according to a systematic review by Drumheller and colleagues, later bedtime and wake-up times and a decrease in sleep duration were found as the screen time use increased [202]. 

### 4.3. Strengths and Limitations

Our study has several strengths. Firstly, to the best of our knowledge, this is the first systematic review and meta-analysis that has sought to provide a comprehensive synthesis of changes in quantitative sleep parameters in the general population during the first lockdown compared to pre-lockdown levels. Despite the abundance of research on the topic, existing systematic reviews or meta-analyses do not offer as comprehensive and detailed coverage of these phenomena as provided by our work. Moreover, our research stands out for its thorough analysis of changes associated with atypical sleep duration and napping habits, areas often overlooked or not fully examined in previous studies. A meticulous methodological approach and an extensive literature review were utilized to explore sleep changes. Additionally, using a random-effect model with a more conservative estimation allowed us to at least partially address the heterogeneity between studies, given the differences in socio-cultural realities, methodological aspects (study design and outcome measures), and the severity of the lockdown measures. The study also has many limitations. The heterogeneity of the collected data limited the subgroup analysis exclusively to the risk of bias and to the countries’ areas. The comparison of the areas is also limited to sleep duration. The results of this study refer exclusively to the first lockdown. Finally, the fact that the lockdown measures differed widely across countries could have affected the study’s outcomes.

### 4.4. Implications of the Results

While the implementation of lockdown measures allowed governments to control the transmission of the SARS-CoV-2 infection, it also had adverse effects on the sleep health of the general population, in particular, on circadian rhythms and sleep–wake cycles. Given the health implications of these changes, and to prepare for possible future pandemics, these findings underline the importance of promoting healthy sleep hygiene, implementing screening programs, and treating sleep disturbances appropriately. However, extending beyond the initial waves of infections, sleep-related issues have persisted in capturing the attention of the scientific community, particularly concerning patients classified as experiencing long COVID. As defined by the WHO, long COVID encompasses a condition emerging in individuals with a history of probable or confirmed SARS-CoV-2 infection. It typically manifests around three months after symptom onset, enduring for at least two months and evading alternative diagnoses. Symptoms may emerge following the initial recovery from acute SARS-CoV-2 infection or persist from the original illness, exhibiting variations and even recurrences over time, often significantly associated with daily functioning. Notably, sleep disturbances are among these symptoms. Consequently, the implications of our findings acquire heightened significance and broad relevance when considered within the broader scope of investigating the intricate complexities of long COVID and its relationship with individuals’ health and well-being. Finally, as there have been other pandemic waves and related lockdowns since that time, it would be essential to uncover whether sleep parameters have returned to pre-pandemic levels or they have become chronic. High-quality, preferably longitudinal, studies are needed to answer these questions.

## 5. Conclusions

This systematic review and meta-analysis uncovered noteworthy changes in quantitative sleep parameters in the general population during the first COVID-19 lockdown. In particular, an increase in sleep duration and a significant delay in sleep timing were uncovered. COVID-19 pandemic affected several basic life aspects of the general population, influencing overall health with implications for both the immune system efficacy and successful vaccination. High-quality research based on longitudinal studies is needed to evaluate the prolonged effects of lockdown on these sleep parameters. At the same time, health authorities and professionals are called upon to address the problem of poor sleep hygiene and to implement intervention strategies as an integral part of overall health management.

## Figures and Tables

**Figure 1 ijerph-21-00583-f001:**
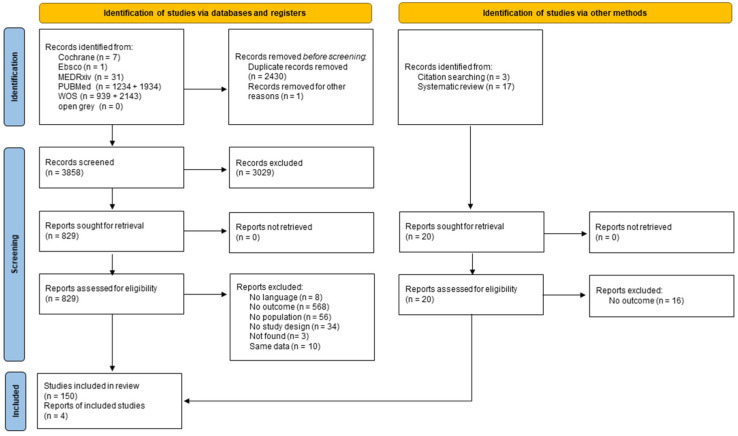
PRISMA flow diagram of study selection.

**Figure 2 ijerph-21-00583-f002:**
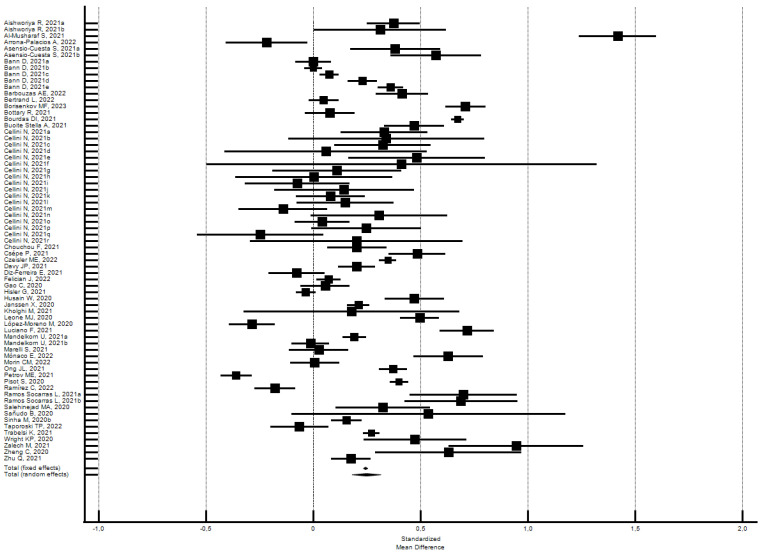
Forest plot showing pooled changes in sleep duration (hours) between before and during the lockdown. Caption: error bars = 95% confidence interval; square boxes = individual study point estimates; diamond box = pooled point estimates. Aishworiya R., 2021a: female; Aishworiya R., 2021b: male; Asensio-Cuesta S., 2021a: male; Asensio-Cuesta S., 2021b: female; Bann D., 2021a: 1946; Bann D., 2021b: 1958; Bann D., 2021c: 1970; Bann D., 2021d: 1990; Bann D., 2021e: 2001; Cellini N., 2021a: Belgian regular workers, female; Cellini N., 2021b: Belgian regular workers, male; Cellini N., 2021c: Belgian remote workers, female; Cellini N., 2021d: Belgian remote workers, male; Cellini N., 2021e: Belgian students, female; Cellini N., 2021f: Belgian students, male; Cellini N., 2021g: Belgian unemployed/retired, female; Cellini N., 2021h: Belgian unemployed/retired, male; Cellini N., 2021i: Italian regular workers, female; Cellini N., 2021j: Italian regular workers, male; Cellini N., 2021k: Italian remote workers, female; Cellini N., 2021l: Italian remote workers, male; Cellini N., 2021m: Italian Stop working, female; Cellini N., 2021n: Italian Stop working, male; Cellini N., 2021o: Italian students, female; Cellini N., 2021p: Italian students, male; Cellini N., 2021q: Italian unemployed/retired, female; Cellini N., 2021r: Italian unemployed/retired, male; Mandelkorn U., 2021a: Study 1; Mandelkorn U., 2021b: Study 2; Ramos Socarras L., 2021a: 18–21-years old; Ramos Socarras L., 2021b: 22–25-years old.

**Figure 3 ijerph-21-00583-f003:**
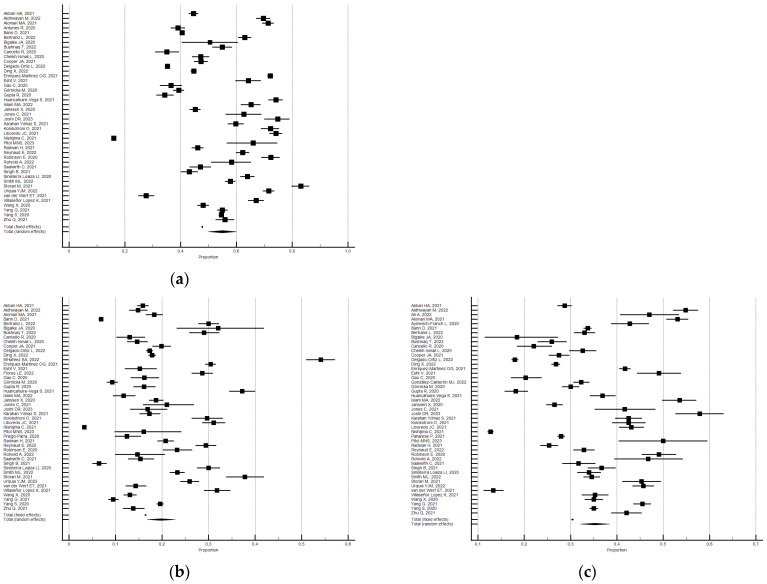
Forest plot showing a pooled percentage of changes in sleep duration from before to during the lockdown: changes (**a**), decrease (**b**), and increase (**c**). Caption: error bars = 95% confidence interval; square boxes = individual point estimates; diamond box = pooled point estimates.

**Figure 4 ijerph-21-00583-f004:**
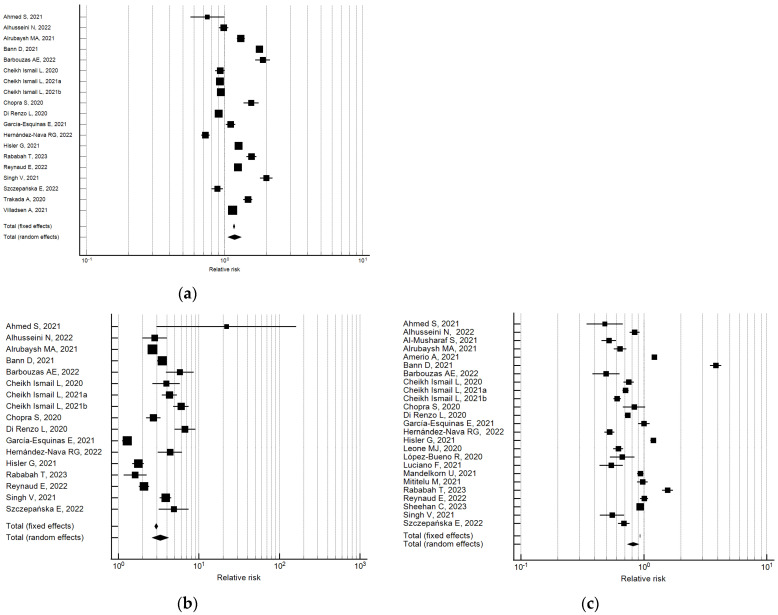
Forest plot showing pooled changes in atypical sleep duration from before to during the lockdown: atypical (**a**), decrease (**b**), increase (**c**). Caption: error bars = 95% confidence interval; square boxes = individual study point estimates; diamond box = pooled point estimates.

**Figure 5 ijerph-21-00583-f005:**
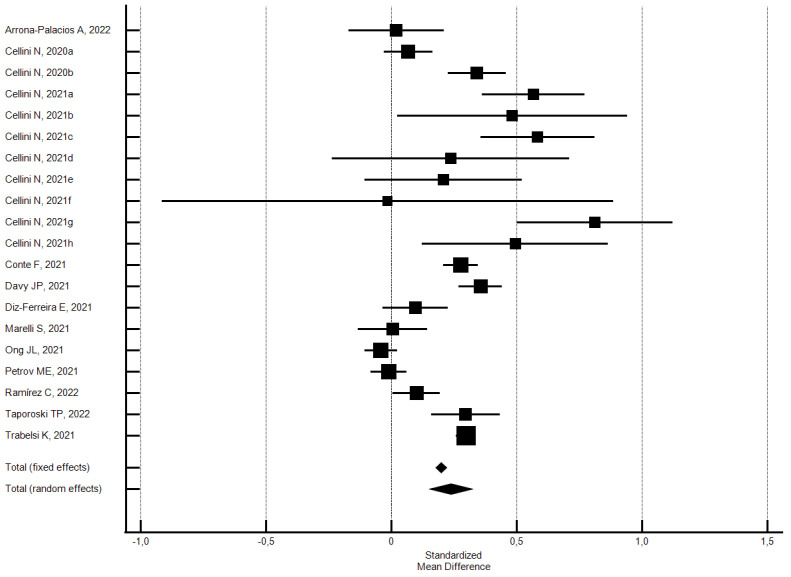
Forest plot showing pooled changes in TIB from before to during the lockdown: increase, decrease. Caption: error bars = 95% confidence interval; square boxes = individual point estimates; diamond box = pooled point estimates. Cellini N., 2020a: student; Cellini N., 2020b: worker; Cellini N., 2021a: Belgian regular workers, female; Cellini N., 2021b: Belgian regular workers, male; Cellini N., 2021c: Belgian remote workers, female; Cellini N., 2021d: Belgian remote workers, male; Cellini N., 2021e: Belgian students, female; Cellini N., 2021f: Belgian students, male; Cellini N., 2021g: Belgian unemployed/retired, female; Cellini N., 2021h: Belgian unemployed/retired, male.

**Figure 6 ijerph-21-00583-f006:**
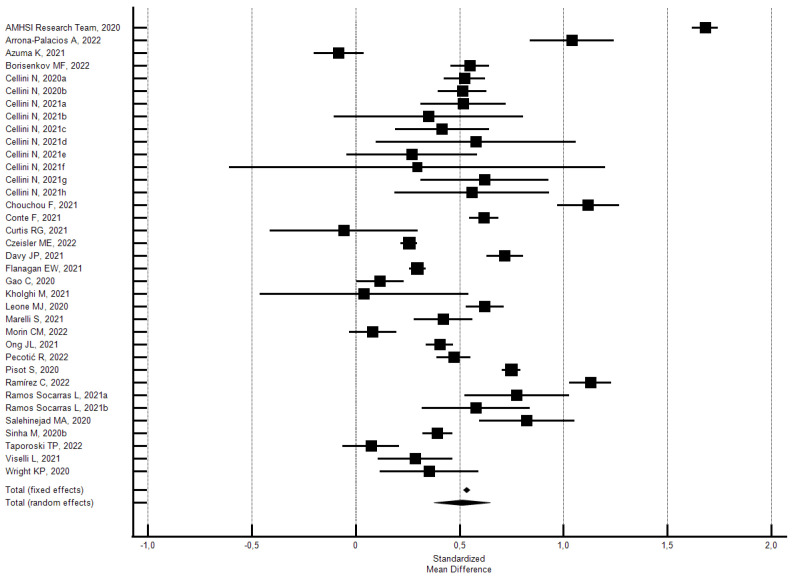
Forest plot showing pooled changes in bedtime from before to during the lockdown. Caption: error bars = 95% confidence interval; square boxes = individual study point estimates; diamond box = pooled point estimates. Cellini N., 2020a: student; Cellini N., 2020b: worker; Cellini N., 2021a: Belgian regular workers, female; Cellini N., 2021b: Belgian regular workers, male; Cellini N., 2021c: Belgian remote workers, female; Cellini N., 2021d: Belgian remote workers, male; Cellini N., 2021e: Belgian students, female; Cellini N., 2021f: Belgian students, male; Cellini N., 2021g: Belgian unemployed/retired, female; Cellini N., 2021h: Belgian unemployed/retired, male; Ramos Socarras L., 2021a: 18–21-years old; Ramos Socarras L., 2021b: 22–25-years old.

**Figure 7 ijerph-21-00583-f007:**
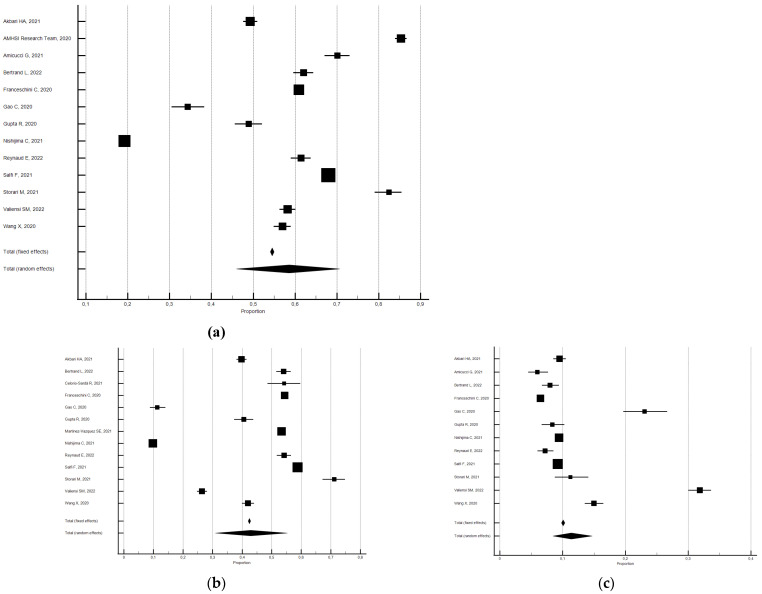
Forest plot showing pooled changes in bedtime from before to during the lockdown: changes (**a**), delayed (**b**), earlier (**c**). Caption: error bars = 95% confidence interval; square boxes = individual point estimates; diamond box = pooled point estimates.

**Figure 8 ijerph-21-00583-f008:**
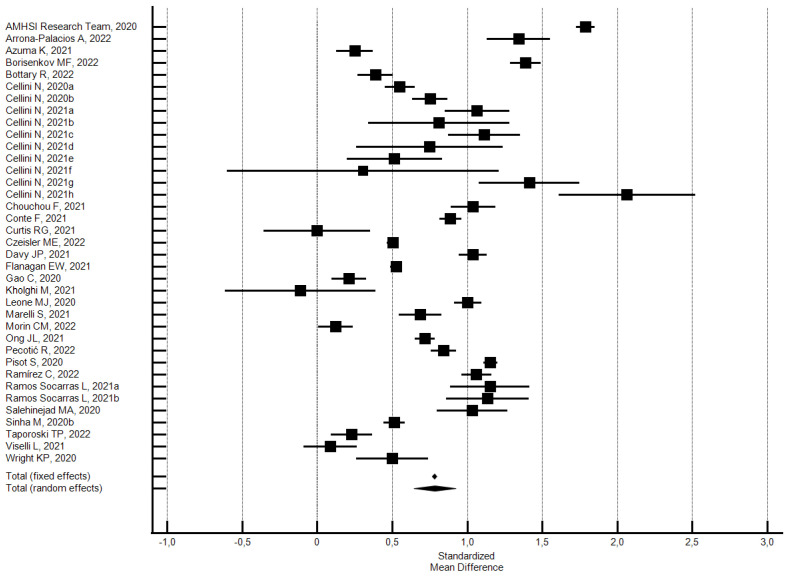
Forest plot showing pooled changes in wake-up time from before to during the lockdown. Caption: error bars = 95% confidence interval; square boxes = individual study point estimates; diamond box = pooled point estimates. Cellini N., 2020a: student; Cellini N., 2020b: worker; Cellini N., 2021a: Belgian regular workers, female; Cellini N., 2021b: Belgian regular workers, male; Cellini N., 2021c: Belgian remote workers, female; Cellini N., 2021d: Belgian remote workers, male; Cellini N., 2021e: Belgian students, female; Cellini N., 2021f: Belgian students, male; Cellini N., 2021g: Belgian unemployed/retired, female; Cellini N., 2021h: Belgian unemployed/retired, male; Ramos Socarras L., 2021a: 18–21-years old; Ramos Socarras L., 2021b: 22–25-years old.

**Figure 9 ijerph-21-00583-f009:**
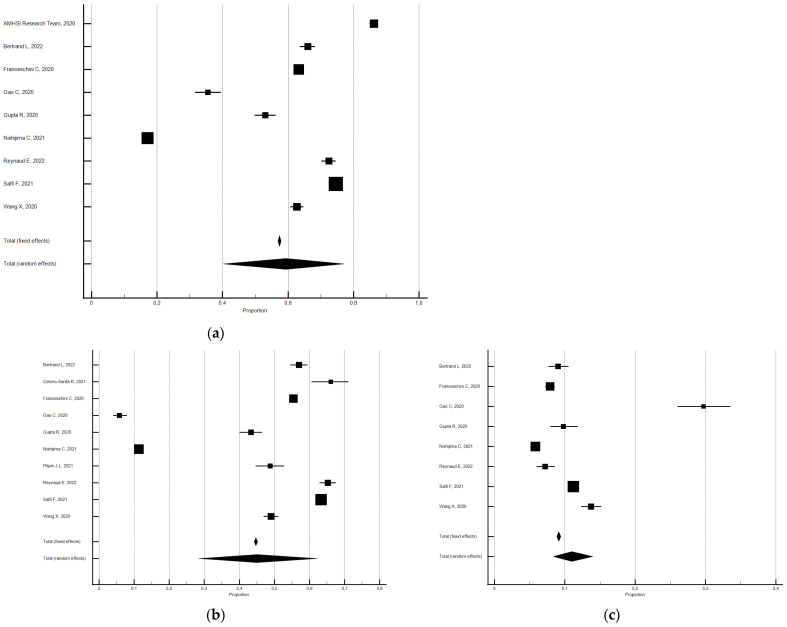
Forest plot showing pooled changes in wake-up time from before to during the lockdown: changes (**a**), delayed (**b**), earlier (**c**). Caption: error bars = 95% confidence interval; square boxes = individual point estimates; diamond box = pooled point estimates.

**Table 1 ijerph-21-00583-t001:** Descriptive characteristics of the studies included.

Author, Year	Country	Study Design	Assessment Period	Population	N (F%)	Age	Data Collection/ Type of Recruitment	Measurement	Risk of Bias Score	D	TIB	BT	WT	NAP
Abouzid M., 2021 [25]	multi-country: Middle East/North Africa countries	cross-sectional	August–4 September 2020	general population	5896; F 62.8%	≥18 y	online survey/snowball sampling	self-report measure	3	D				
Ahmed S., 2021 [26]	Bangladesh	cross-sectional	24 April–25 May 2020	general population	230; F 20.9%	8–60 y	online survey/random sampling	self-report measure	4	D				
Aishworiya R., 2021 [27]	Singapore	cross-sectional	7 April–1 June 2020	general population	593; F 86.0%	≥21 y	online survey/convenience sampling	self-report measure	6	D		BT	WT	
Akbari H.A., 2021 [28]	Iran	cross-sectional	17 November 2020–13 February 2021	general population	3323; F 54.3%	30 ± 11 y	online survey/snowball sampling	self-report measure	5	D		BT		
Aldhwayan M., 2022 [29]	Saudi Arabia	cross-sectional	2–23 April 2020	general population	1860; F 75.1%	>18; median 36 y (IQR 18)	online survey/convenience sampling	self-report measure	4	D				
Alhusseini N., 2022 [30]	Saudi Arabia	cross-sectional	22 May–2 June 2021	general population	1051; F 71%	≥18 y	online survey/convenience sampling	self-report measure	3	D				
Ali A., 2021 [31]	Pakistan	cross-sectional	24 March–26 April 2020	students	251; F 70.2%	19.4 ± 1.6 y	online survey/convenience sampling	self-report measure	6	D				
Al-Musharaf S., 2021 [32]	Saudi Arabia	longitudinal	B: February–April 2019; D: April–May 2020	university students	297; F 100%	19–30 y; 20.7 ± 1.4 y	telephone interview/nr	self-report measure	6	D				
Alomari M.A., 2021 [33]	Jordan	cross-sectional	second-third quartiles of 2020	general population	1757; F 69.4%	33.8 ±11.1y; ≥18 y	online survey/convenience sampling	self-report measure	4	D				NAP
Alrubaysh M.A., 2021 [34]	Saudi Arabia	cross-sectional	January 2021–February 2021	general population	2069; F 68.1%	≥18 y	online survey/convenience sampling	self-report measure	3	D				
Amerio A., 2021 [35]	Italy	cross-sectional	27 April–3 May 2020	general population	6003; F 50,7%	18–74 y	online survey/quota sampling method	self-report measure	6	D				
AMHSI Research Team, 2021 [36]	multi-country: International	longitudinal	D: 1 March–15 June 2020	adults	2645; F 52.5%	19–60 y	online and telephone survey/snowball sampling	self-report measure	5			BT	WT	NAP
Anastasiou E., 2021 [37]	Greece	cross-sectional	31 March–23 April 2020	general population	4216; F 70,87%	36.8 ± 12.0 y	online survey/convenience sampling	self-report measure	7			BT	WT	
Antunes R., 2020 [38]	Portugal	cross-sectional	1–15 April 2020	general population	1404; F 69.6%	18–89 y, 36.4 ± 11.7 y	online survey/convenience sampling	self-report measure	6	D				
Arrona-Palacios A., 2022 [39]	Mexico	cross-sectional	18 May–10 June 2020	faculty members of universities	214; F 56.5%	42.66 ± 9.17 y; 25–64 y	online survey/snowball sampling	self-report measure	4	D	TIB	BT	WT	
Asensio-Cuesta S., 2021 [40]	Spain	cross-sectional	B: 17 October 2019–17 February 2020; D: 21 April–1 May 2020	university community (students, teachers, and staff)	B: 341, F 43.1%; D: 398, F 58.3%	≥18 y	telegram chatbot/convenience sampling	self-report measure	5	D				
Aymerich-Franch L., 2020 [41]	Spain	cross-sectional	15–25 April 2020	general population	584; F 75.3%	18–65 y	online survey/snowball sampling	self-report measure	2	D				
Azizi A., 2020 [42]	Morocco	cross-sectional	B: nr; D: 9–30 May 2020	general population	B: 484, F nr; D: 537, F 62.9%	D: 33.19 ± 12.14	online survey/convenience sampling	self-report measure	4			BT	WT	NAP
Azuma K., 2021 [43]	Japan	cross-sectional	B: January 7–28 April 2019; D: 6 January 6–26 April 2020	general population	B: 464, F 74.6%; D: 622, F 85.7%	≥20 y; B 35 ± 12 y; D: 32 ± 11 y	online survey/convenience sampling	self-report measure	6			BT	WT	
Bann D., 2021 [44]	UK	cross-sectional	4–30 May 2020	general population by birth cohort	13283; F 50.3%	≥19 y	online survey/cohort	self-report measure	6	D				
Barbouzas A.E., 2022 [45]	Greece	cross-sectional	2 September–27 November 2020	young adults	540; F 62.8%	21.2 ± 2.3 y; 18–25 y	online survey/cohort	self-report measure	4	D				
Bertrand L., 2022 [46]	France	cross-sectional	6–11 May 2020	general population	1627; F 74.3%	<18 y 1%–>65 y 7.5%	online survey/convenience sampling	self-report measure	4	D		BT	WT	
Bigalke J.A., 2020 [47]	USA	cross-sectional	25 April–18 May 2020	general population	103; F 59%	mean 38 y	online survey/convenience sampling	self-report measure	5	D				
Blume C., 2021 [48]	multi-country: Austria/Germany/Switzerland	cross-sectional	23 March–26 April 2020	general population	435; F 75.2%	≥18 y	online survey/convenience sampling	self-report measure	6	D				
Borisenkov M.F., 2022 [49]	Russia	cross-sectional	17 April–14 June 2020	university students	B: 1050, F 71.8; D: 844; F 79.4%	B: 18.9 ± 1.9; D: 19.4 ± 1.8 y	online survey/convenience sampling	self-report measure	5	D		BT	WT	
Bottary R., 2022 [50]	USA	cross-sectional	April–May 2020	general population	610; F 82.9%	≥18 y; 39.24 ± 17.45 y; 18–89 y	online survey/snowball and convenience sampling	self-report measure	5	D		BT	WT	
Bourdas D.I., 2021 [51]	Greece	cross-sectional	4–19 April 2020	general population	8495; F 61.68%	≥18 y	online survey/snowball sampling	self-report measure	5	D				
Buoite Stella A., 2021 [52]	Italy	cross-sectional	23–29 March 2020	general population	400; F 69%	35 ± 15 y	device data or online survey/convenience sampling	objective or self-report measures	6	D				
Bushnaq T., 2022 [53]	Saudi Arabia	cross-sectional	10August–9 October 2021	general population	786; F 88.3%	≥18 y; 30.48 ± 11.50 y	online survey/convenience sampling	self-report measure	4	D				
Cancello R., 2020 [54]	Italy	cross-sectional	15 April–4 May 2020	general population	490; F 84%	≥18 y	online survey/convenience sampling	self-report measure	2	D				
Casas R., 2022 [55]	Spain	cross-sectional	23 April–2 June 2020	general population	945; F 70.8%	>18 y; 43.4 ± 13.4 y	online survey/snowball sampling	self-report measure	3	D				
Cellini N., 2020 [56]	Italy	cross-sectional	24–28 March 2020	students/workers	1310; F 67.2%	23.91 ± 3.6 y	online survey/convenience sampling	self-report measure	5		TIB	BT	WT	
Cellini N., 2021 [57]	multi-country: Italy/Belgium	cross-sectional	1 April–19 May 2020	general population	1622 Italians, F 72.2%; 650 Belgian, F 78.3%	34.1 ± 13.6 y; 43.0 ± 16.8 y	online survey/convenience sampling	self-report measure	5	D	TIB	BT	WT	
Celorio-Sardà R., 2021 [58]	Spain	cross-sectional	22 May–3 July 2020	students/workers	321; F 79.8%	≥18 y	online survey/convenience sampling	self-report measure	3			BT	WT	
Cheikh Ismail L., 2021b [59]	Lebanon	cross-sectional	3–28 June 2020	general population	2507; F 73%	>18 y	online survey/snowball sampling	self-report measure	3	D				
Cheikh Ismail L., 2020 [60]	United Arab Emirates	cross-sectional	April–May 2020	general population	1012; F 75.9%	≥18 y	online survey/convenience sampling	self-report measure	5	D				
Cheikh Ismail L., 2021a [61]	multi-country MENA region: Algeria, Bahrain, Egypt, Iraq, Jordan, Kuwait, Lebanon, Libya, Morocco, Oman, Palestine, Qatar, Saudi Arabia, Sudan, Syria, Tunisia, United Arab Emirates and Yemen.	Cross-sectional	15–29 April 2020	general population	2970; F 71·6%	≥18 y	online survey/convenience and snowball sampling	self-report measure	3	D				
Chopra S., 2020 [62]	India	cross-sectional	15–30 August 2020	general population	995; F 41.4%	33.33 ± 14.5 y; 18–85 y	online and telephone survey/quota sampling	self-report measure	3	D				
Chouchou F., 2021 [63]	France	cross-sectional	35–45th days of lookdown	general population	400; F 58.3%	≥18 y; 29.8 ± 11.5 y	online survey/convenience sampling	self-report measure	4	D		BT	WT	NAP
Conte F., 2021 [64]	Italy	cross-sectional	1–20 April 2020	general population	1622; F 72.2%	>18 y; 18–79 y; 34.1 ± 13.6 y	online survey/convenience sampling	self-report measure	5		TIB	BT	WT	NAP
Cooper J.A., 2021 [65]	USA	cross-sectional	24 April–4 May 2020	adults	1607; F 56.6%	38.0 ± 12.9 y, 18–75 y	online survey/convenience sampling	self-report measure	5	D				
Csépe P., 2021 [66]	Hungary	cross-sectional	29 June–5 July 2020	university students	447; F 75,16%	25.6 ± 7.2 y	online survey/convenience sampling	self-report measure	4	D				
Curtis R.G., 2021 [67]	Australia	longitudinal	B: 10–23 February 2020; D: 14–27 April 2020	parents of children	61; F 66%	41 ± 6 y	device data/sample of parents	objective measure (wrist-worn wearable device, Fitbit Charge 3)	6			BT	WT	
Czeisler M.E., 2022 [68]	USA	longitudinal	B: 1 January–12 March 2020; D: 13 March–12 April 2020	general population	4912; F 29.3%	≥18 y	device data/non-probability sampling	objective measure (sleep wearable device, WHOOP)	6	D		BT	WT	
Czeisler M.É., 2021 [69]	Australia	cross-sectional	15–24 September 2020	general population	1157; F 53%	≥18 y	online survey/demographic quota sampling and survey weighting to Census	self-report measure	4		TIB			
Davy J.P., 2021 [70]	South Africa	cross-sectional	12 May–15 June 2020	young adults	1048; F 73.2%	median 27 (21, 42) y	online survey/convenience and snowball sampling	self-report measure	6	D	TIB	BT	WT	
Delgado-Ortiz L., 2022 [71]	Catalonia	cross-sectional	March–August 2020	general population	10032; F 59%	55.3 ± 8 y	online or telephone survey/cohort	self-report measure	8	D				
Di Renzo L., 2020 [72]	Italy	cross-sectional	5–24 April 2020	general population Internet users	3533; F 76.1%	40.03 ± 13.53 y	online survey/convenience sampling	self-report measure	4	D				
Ding X., 2022 [73]	UK	cross-sectional	2–31 May 2020	general population	8547; F 57%	≥17 y	online survey/cohort	self-report measure	6	D				
Diz-Ferreira E., 2021 [74]	Spain	cross-sectional	30 March–12 April 2020	general population	451; F 73.4%	≥18 y	paper and online survey/convenience sampling	self-report measure	4	D	TIB		WT	
Dragun R., 2021 [75]	Croatia	cross-sectional	B: 2018–2019; D: May 2020	students	B: 1326, F 63.8%; D: 531, F 62.3%	B/D: 18.0 (IQR 6.0)	online survey/convenience sampling	self-report measure	5	D				
Elhadi M., 2021 [76]	Libya	cross-sectional	18 July–23 August 2020	general population	10,296; F 76.6%	28.9 ± 8.5 y	online survey/convenience sampling	self-report measure	3			BT	WT	
ElHafeez S.A., 2022 [77]	Egypt	cross-sectional	25 April–1 June 2020	general population	1000; F 66.2%	≥18 y	online survey/convenience sampling	self-report measure	5	D				
Enriquez-Martinez O.G., 2021 [78]	multi-country: Argentina, Brazil, Mexico, Peru, Spain	cross-sectional	1 April–26 September 2020	general population	6325; F 68.1%	≥18 y	online survey/convenience sampling	self-report measure	5	D				
Esht V., 2021 [79]	India	cross-sectional	April–May 2020	general population	440; F 52.7%	20–40 y	online survey/snowball sampling	self-report measure	4	D				
Falkingham J., 2020 [80]	UK	longitudinal	B: 2019; D: April 2020	general population	8163; F 53%	50.6 ± 17.5 y	online survey/nr	self-report measure	7	D				
Felician J., 2022 [81]	France	cross-sectional	9 April–9 June 2020	general population	2513; F 77%	median 39 (IQR 30–48) y	online survey/convenience sampling	self-report measure	7	D				NAP
Flanagan E.W., 2021 [82]	multi-country: USA/Australia/Canada/Ireland/UK	cross-sectional	3 April–3 May 2020	general population	7753; F 80.0%	≥18 y; 51.2 ± 0.17 y	online survey/convenience sampling	self-report measure	5			BT	WT	
Flores L.E., 2022 [83]	Argentina	cross-sectional	2–22 December 2020	general population	1536; F 75.1%	≥18 y; 38.8 ± 13.1 y;	online survey/convenience sampling	self-report measure	5	D				
Franceschini C., 2020 [84]	Italy	cross-sectional	March 10–4 May 2020	adults	6439; F 73.1%	33.9 ± 27.6 y; 18–82	online survey/convenience sampling	self-report measure	5			BT	WT	NAP
Gao C., 2020 [85]	USA	longitudinal	B: 17 February 2020; D: 25–27 March 2020	general population	699 (B:199; D: 500; B/D: 86); F 44.78%	38.04 ± 11.65 y	online survey/convenience sampling	self-report measure	5	D		BT	WT	
García-Esquinas E., 2021 [86]	Spain	longitudinal	B: 2019; D: 27 April–22 June 2020	general population	3041; F 57.7%	≥65 y	telephone interview/cohort	self-report measure	6	D				NAP
García-Garro P.A., 2022 [87]	Colombia	cross-sectional	26 May–23 June 2020	university community (professors, administration staff)	354; F 40.96%	43.39 ± 10.21 y; 40–64 y	online survey/convenience sampling (calculation of the sample size)	self-report measure	5	D				
Gibson R., 2022 [88]	New Zealand	cross-sectional	11 April 2020	general population	723; F 82.3%	median 45 (IQR 22) y; 20–85 y	online survey/convenience sampling	self-report measure	5	D	TIB	BT	WT	
González-Calderón M.J., 2022 [89]	Spain	cross-sectional	9–31 May 2020	university community (students, professors, administration staff)	2834; F 69.3%	41.36 ± 10.5 y; 19–76 y	online survey/convenience sampling	self-report measure	4	D				
Gornicka M., 2020 [90]	Poland	cross-sectional	30 April–23 May 2020	general population	2381; F 89.8%	≥18 y	online survey/convenience sampling	self-report measure	5	D				
Gupta R., 2020 [91]	India	cross-sectional	28 April–10 May 2020	general population	958; F 41.2%	37.32 ± 13.09 y	online survey/snowball sampling	self-report measure	5	D		BT	WT	NAP
Hernández-Nava R.G., 2022 [92]	Mexico	cross-sectional	2 June–4 July 2020	general population	1004; F 69.5%	≥18 y	online survey/convenience sampling	self-report measure	5	D				
Hisler G., 2021 [93]	USA	longitudinal	B: 2018; D: 27 April 2020	general population	B: 19433, F 51.8%; D: 2059, F 50.7%	≥18 y; B: 42.84 ± 14.84 y; D: 43.35 ± 14.88 y	online survey/probability sampling	self-report measure	7	D				
Huancahuire-Vega S., 2021 [94]	Peru	cross-sectional	16 July–31 August 2020	general population	1176; F 51.5%	≥18 y	online survey/convenience sampling	self-report measure	5	D				
Husain W., 2020 [95]	Kuwait	cross-sectional	30 March–15 April 2020	adults	415; F 68.7%	≥18 y; 38.47 ± 12.73 y, 18–73 y	online survey/snowball sampling	self-report measure	4	D				
Islam M.A., 2022 [96]	Bangladesh	cross-sectional	10–17 December 2020	general population	748; F 41.3%	≥18 y	online survey/snowball sampling	self-report measure	4	D				
Janssen X., 2020 [97]	UK: Scotland	longitudinal	D: 20 May–12 June 2020	general population	3230; F 79.2%	≥18 y, 46.2 ± 15.3 y	online survey/convenience sampling	self-report measure	5	D				
Jones C., 2021 [98]	USA	cross-sectional	14 May–24 October 2020	general population	228; F 79.0%	≥18 y; 45.0 ± 17.1 y	online survey/convenience sampling	self-report measure	3	D				
Joshi D.R., 2023 [99]	Nepal	cross-sectional	15 April–25 July 2020	academicians (school teachers, faculty members, and graduate students of higher education institutions)	361; F 18.3%	34.17 ± 8.67 y	online survey/snowball sampling	self-report measure	6	D				
Kaizi-Lutu M., 2021 [100]	USA	cross-sectional	16 May–11 November 2020	general population	226; F 77.8%	≥18 y; 44.9 ± 17.4 y	online survey/convenience sampling	self-report measure	3			BT	WT	NAP
Karahan Yılmaz S., 2020 [101]	Turkey	cross-sectional	April–May 2020	adults	1120; F 63.2%	18–65 y; 33.04 ± 11.04 y	online survey/convenience sampling	self-report measure	2	D				
Khojasteh M.R., 2022 [102]	Iran	cross-sectional	March–April 2020	university students	283; F 72.4%	24.11 ± 2.54	online survey/convenience sampling	self-report measure	5	D				
Kholghi M., 2021 [103]	Australia	longitudinal	B: November 2019–February 2020; D: March–May 2020	older adults	31; 54.8%	84 ± 6.8 y	device data/cohort	objective measure (mattress-based devices, EMFIT QS)	7	D		BT	WT	
Kim A.C.H., 2022 [104]	USA	cross-sectional	first week of June 2020	general population	695; F 40%	45.85 ± 15.42 y	online survey/convenience sampling	self-report measure	4	D				
Kolokotroni O., 2021 [105]	Cyprus	cross-sectional	10 April–12 May 2020	general population	745; F 73.8%	≥18 y; median 39 (IQR 13) y; 18–76 y	online survey/convenience sampling	self-report measure	4	D				
Kontsevaya A.V., 2021 [106]	Russia	cross-sectional	26 April–6 June 2020	general population	2432; F 83%	≥18 y; 37.6 ± 13.4 y	online survey/convenience sampling	self-report measure	5	D			WT	
Leone M.J., 2020 [107]	Argentina	longitudinal	B: February and May 2018 and 2019/February 2020; D: April 2020	general population	1021; F 69.64%	13–74 y; 37.4 ± 13.21 y	online survey/convenience sampling	self-report measure	7	D		BT	WT	NAP
Li J.W., 2021 [108]	China	longitudinal	B: 26 December 2019-22 January 2020; D: 3–21 January 21 February 2020	general population	19,960; F 10.1%	35.7 ± 11.3 y	device data/cohort	objective measure (wrist-worn wearable device—acceleration sensor and photoplethysmogram)	6	D				
Liboredo J.C., 2021 [109]	Brazil	cross-sectional	14 August–9 September 2020	general population	1368; F 80%	median 31 y; 18–87 y	online survey/convenience sampling	self-report measure	4	D				
Lopez-Bueno R., 2020 [110]	Spain	cross-sectional	22 March–5 April 2020	adults	2741; F 51.8%	≥18 y; 34.2 ± 13.0 y	online survey/convenience sampling	self-report measure	6	D				
López-Moreno M., 2020 [111]	Spain	cross-sectional	28 May–21 June 2020	general population	675; F 30.1%	≥18 y; 39.1 ± 12.9 y; 18–85 y	online survey/snowball sampling	self-report measure	3	D				
Luciano F., 2020 [112]	Italy	cross-sectional	B: October–November 2019; D: 9 March–3 May 2020	university students	B: 714, F 62%; D: 394, F 73%	B/D: 25 ± 2 y	online survey/convenience sampling	self-report measure	4	D				
Majumdar P., 2020 [113]	India	cross-sectional	14 April–2 May 2020	university students/workers	325 students, F 60.9%; 203 workers, F 18.2%	33.1 ± 7.11 y; 22.1 ± 1.66 y	online survey/convenience sampling	self-report measure	3	D				NAP
Mandelkorn U., 2021 [114]	multi-country: multi-national/USA	cross-sectional	26 March–26 April 2020	general population	2562 study 1, F 68%; 971 study 2, F 52.8%	study 1 45.18 ± 14.46 y; study 2 40.36 ± 13.61 y	online survey/convenience sampling	self-report measure	2	D				
Marelli S., 2021 [115]	Italy	cross-sectional	24 March–2 May 2020	university students/workers	400; F 75.8%	22.84 ± 2.68 y	online survey/convenience sampling	self-report measure	6	D	TIB	BT	WT	
Martínez-Vázquez S.E., 2021 [116]	Mexico	cross-sectional	13 April–16 May 2020	general population	8289; F 80%	≥18 y; 18–38	online survey/snowball sampling	self-report measure	4			BT		
Mititelu M., 2021 [117]	Romania	cross-sectional	8–26 July 2020	general population	805, F 19.7%	≥20 y	online survey/convenience sampling	self-report measure	4	D				
Mohsin A., 2021 [118]	Pakistan	cross-sectional	27 May–1 July 2020	general population	553; F 63.5%	>18 y	online survey/convenience sampling	self-report measure	3					NAP
Mónaco E., 2022 [119]	Spain	longitudinal	30 March 2020	general population	B: 363, F 69.4%; D: 261, F nr	32.59 ± 12.57 y; age range: 18–65 y	online survey/snowball sampling	self-report measure	3	D		BT	WT	NAP
Morin C.M., 2022 [120]	Canada	longitudinal	B: 2018; D: April–May 2020	general population	594; F: 64.0%	48.3 ± 13.1 y; 18–83 y	online or telephone survey/cohort	self-report measure	5	D		BT	WT	NAP
Nishijima C., 2021 [121]	Japan	cross-sectional	9–14 September 2020	general population	9645; F 52,4%	≥20 y	online survey/random sampling by age, sex, and place of residence	self-report measure	5	D		BT	WT	
Ong J.L., 2021 [122]	Singapore	longitudinal	B: 2–22 January 2020; D: 7–27 April 2020	city-dwelling/young working adults	1824; F 51.64%	21–40 y; 30.94 ± 4.62 y	device data/convenience sampling	objective measure (wrist-worn wearable device, Fitbit API)	9	D	TIB	BT	WT	
Pachocka L., 2022 [123]	Poland	cross-sectional	August 2020	general population	490; F 66.1%	18–80 y	face to face survey/convenience sampling	self-report measure	3	D				
Panarese P., 2021 [124]	Italy	cross-sectional	7 April–3 May 2020	general population	11,452; F nr	≥25 y	online survey/snowball sampling	self-report measure	3	D				
Pecotić R., 2022 [125]	Croatia	cross-sectional	25 April–5 May 2020	general population	1173; F 73.7%	≥18 y; median 42 (32–52) y	online survey/snowball sampling	self-report measure	6			BT	WT	
Pépin J.-L., 2021 [126]	France	longitudinal	B: 16 February–March 2020; D: March 17–11 May 2020	regular users of a sleep-monitoring headband	599; F 29%	median 47 (IQR 36–59) y	device data or online survey/convenience sampling	objective (dream sleep-monitoring headband) and self-report measures	8	D	TIB	BT	WT	
Perez-Carbonell L., 2020 [127]	UK	cross-sectional	12 May–2 June 2020	general population	843; F 67.4%	≥18 y; median 52 (IQR 40–63) y	online survey/convenience sampling	self-report measure	2			BT		
Peterson M., 2021 [128]	USA	longitudinal	B: before 15 March 2020; D: after 15 March 2020	general population	9; F 55.6%	22–48 y	device data/nr	objective measure (wrist-worn actigraph, Actiwatch-2 + non-contact monitoring device, SleepScore Max, SleepScore Labs)	5	D				
Petrov M.E., 2021 [129]	multi-country 79 countries	cross-sectional	21 May 2020–7 July 2020	general population	991; 72.5%	≥18; 37.9 ± 14.6 y; 18–80 y	online survey/convenience sampling	self-report measure	5	D	TIB			NAP
Pisot S., 2020 [130]	multi-country: Bosnia and Herzegovina/Croatia/Greece/Kosovo/Italy/Serbia/Slovakia/Slovenia/Spain	cross-sectional	15 April–3 May 2020	general population	4108; F 63.6%	15–82 y; 32.0 ± 13.2 y	online survey/snowball sampling	self-report measure	5	D		BT	WT	
Pitol M.N.S., 2023 [131]	Malaysia	cross-sectional	first lockdown	general population	112; F 68.8%	≥18 y; 19–60 y	online survey/convenience sampling	self-report measure	4	D				
Pouget M., 2022 [132]	France	cross-sectional	26 June 2020–2 March 2021	general population	671; F 74%	47 ± 13 y	online survey/convenience sampling	self-report measure	4	D				
Priego-Parra, 2020 [133]	Mexico	cross-sectional	23 March–21 April 2020	general population	561; F 71%	30.7 ± 10.6 y	online survey/snowball sampling	self-report measure	6	D				
Rababah T., 2023 [134]	Jordan	cross-sectional	March–June 2021	general population	672; F61.9%	≥18 y	online survey/convenience sampling	self-report measure	3	D				
Radwan H., 2021 [135]	United Arab Emirates	cross-sectional	5–18 May 2020	adults residing	2060; F 75.1%	≥18 y	online survey/convenience sampling	self-report measure	6	D				
Ramírez C., 2022 [136]	Mexico	cross-sectional	30 April –23 May 2020	general population	861; F 74.7%	18–69 y; 27.73 ± 11.31 y	online survey/snowball sampling	self-report measure	5	D	TIB	BT	WT	NAP
Ramos Socarras, 2021 [137]	Canada	cross-sectional	3 June–3 July 2020.	young adults	248; F 75.4%	18–25 y	online survey/convenience sampling	self-report measure	6	D		BT	WT	
Reynaud E., 2022 [138]	France	cross-sectional	11 April –20 May 2020	general population	1652; F 77.1%	≥18 y; 35.4 ± 11.4 y	online survey/convenience sampling	self-report measure	3	D		BT	WT	
Robinson E., 2020 [139]	UK	cross-sectional	19–22 April 2020	adults	723; F 67%	18–60 y, 30.7 ± 9.6 y	online survey/convenience sampling	self-report measure	3	D				
Rotvold A., 2022 [140]	USA	cross-sectional	spring of 2020	students	195; F74.5%	18–46 y	online survey/convenience sampling	self-report measure	3	D				
Ruiz-Zaldibar C., 2022 [141]	Spain	cross-sectional	11–25 April 2020	university students	488; F 73.6%	median 21 y; 18–54 y	online survey/convenience sampling	self-report measure	6	D				
Saalwirth C., 2021 [142]	Germany	cross-sectional	1–19 April 2020	general population	665; F 53.8%	18–73 y; 36 ± 14 y	online survey/convenience sampling	self-report measure	4	D				
Salehinejad M.A., 2020 [143]	Germany	cross-sectional	20–28 April 2020	general population	160; F 85.6%	18–60 y; 25.79 ± 7.31 y	online survey/convenience sampling	self-report measure	4	D		BT	WT	
Salfi F., 2021 [144]	Italy	cross-sectional	25 March –3 May 2020	general population	13,989; F 76,96%	34.8 ± 12.2 y; 18–86 y	online survey/snowball sampling	self-report measure	4	D		BT	WT	NAP
Santos-Miranda E., 2021 [145]	Spain	cross-sectional	23 March –6 April 2020	general population	474; F 54.9%	31.9 ± 12.1 y; median 29 (IQR 22–41) y	online survey/convenience sampling	self-report measure	3	D				NAP
Sañudo B., 2020 [146]	Spain	longitudinal	B: February 2020; D: 24 March –3 April 2020	general population	20; F 45%	22.6 ± 3.4 y	device data/convenience sampling	objective measure (wristband accelerometer, Xiaomi Mi Band 2)	5	D		BT	WT	
Scarpelli S., 2021 [147]	Italy	cross-sectional	10 March–4 May 2020	general population	5988; F 73.3%	≥18	online survey/convenience sampling	self-report measure	3					NAP
Shahzadi K., 2021 [148]	Pakistan	cross-sectional	1 June–30 July 2020	general population	100; F 68%	18–50 y	online survey/convenience sampling	self-report measure	2				WT	
Sheehan C., 2023 [149]	USA	cross-sectional	B: March 2018; D: March 2020	general population	2,203,861; F 51.2%	≥18 y	telephone survey/random sampling	self-report measure	6	D				
Singh B., 2021 [150]	India	cross-sectional	11–20 May 2020	adults	1008; F 43.4%	18–81 y, median 24 y	online survey/convenience sampling	self-report measure	4	D				
Singh V., 2021 [151]	India	cross-sectional	1–15 June 2020	general population	1251; F 29.5%	31.71 ± 13.5 y	online survey/convenience sampling	self-report measure	2	D				
Sinha M., 2020a [152]	India	cross-sectional	1 April–6 May 2020	general population	1511; F 50.9%	≥18 y; 18–80 y	online survey/convenience sampling	self-report measure	4					NAP
Sinha M., 2020b [153]	India	cross-sectional	1–7 May 2020	general population/university students	1511; F 50.9%	≥18 y	online survey/convenience sampling	self-report measure	6	D		BT	WT	
Sinisterra Loaiza L.I., 2020 [154]	Spain: Galicia	cross-sectional	2–15 May 2020	adults	1350; F 70%	63.2 ± 8.1 y	online survey/convenience sampling	self-report measure	3	D				
Smith M.L., 2022 [155]	UK	cross-sectional	26 May –5 July 2020	young adults	2710; F nr	mean 27.8 y	online survey/cohort	self-report measure	4	D				
Souza T.C., 2022 [156]	Brazil	cross-sectional	August–September 2020	general population	1368; F 80%	≥18 y; median 31 (24–39) y	online survey/convenience sampling	self-report measure	4	D		BT	WT	
Storari M., 2021 [157]	Italy	cross-sectional	29 April –17 May 2020	general population	967; F 58.84%	≥18 y	online survey/convenience sampling	self-report measure	6	D		BT		
Szczepańska E., 2022 [158]	Poland	cross-sectional	2 first weeks of May 2020	parents of children	1098; F nr	20–50 y	online survey/convenience sampling	self-report measure	2	D				
Tang N.K.Y., 2022 [159]	UK	cross-sectional	July–September 2020	university students/young adults	1442; 56.2%	18–30 y	online survey/convenience sampling	self-report measure	6	D		BT		
Taporoski T.P., 2022 [160]	Brazil	longitudinal	B: January 2010–September 2014; D: March 30–29 June 2020	general population	417; F 70%	44 ± 15 y	telephone survey/cohort	self-report measure	7	D	TIB	BT	WT	
Trabelsi K., 2021 [161]	multi-country: Western Asia/North Africa/Europe/Americas	cross-sectional	6 April –28 June 2020	general population	5056; F 59.4%	≥18 y	online survey/convenience sampling	self-report measure	4	D	TIB			
Trakada A., 2020 [162]	multi-country: Greece/Switzerland/Austria/Germany/France/Brazil	cross-sectional	25 March–6 April 2020 (Europe); 10–14 2020 (Brazil)	general population	1622; F nr	nr	online survey/convenience sampling	self-report measure	5	D				
Tsigkas G., 2021 [163]	Greece	cross-sectional	13–30 April 2020	general population	1014; F 48.7%	≥35 y	telephone survey/representative sample	self-report measure	5	D				
Urquia Y.J.M., 2022 [164]	Brazil	cross-sectional	July–September 2020	general population	1828; F 70.5%	18–83 y	online survey/convenience sampling	self-report measure	4	D				
Valiensi S.M., 2022 [165]	Argentina	cross-sectional	13–30 April 2020	general population	2594; F 69%	42 ± 13 y; 18–85 y	online survey/convenience sampling	self-report measure	4			BT		NAP
van der Werf E.T., 2021 [166]	Netherlands	cross-sectional	22–27 May 2020	general population	1004; F 50.7%	18–88 y	online survey/convenience sampling	self-report measure	5	D				
Villadsen A., 2020 [167]	UK	longitudinal	D: May 2020	general population by birth cohort	10666; F 60.4%	19–62 y	online survey/cohort	self-report measure	8	D				
Villasenor Lopez K., 2021 [168]	Mexico	cross-sectional	27 April–17 May 2020	general population	1084; F 66.5%	35.5 ± 13.9 y, 18–86 y	online survey/convenience sampling	self-report measure	5	D				
Vinogradov O.O., 2022 [169]	Ukraine	cross-sectional	10–12 May 2020	university students	86; F 58.1%	22.9 ± 0.56 y	online survey/convenience sampling	self-report measure	2	D		BT	WT	NAP
Viselli L., 2021 [170]	Italy	cross-sectional	B: 6–11 October 2016; D: 25–31 March 2020	university students	B: 240, F 80.42%; D: 240	B/D: 20.39 ± 1.42 y; 18–25 y	nr/non-probability sampling	self-report measure	4			BT	WT	
Vollmer C., 2022 [171]	Austria	cross-sectional	24 April –8 May 2020	teachers	2314; F 72.9%	45.3 ± 10.9 y	online survey/convenience sampling	self-report measure	6	D		BT	WT	
Wang X., 2020 [172]	China	cross-sectional	23 March –26 April 2020	general population	2289; F 48.6%	27.5 ± 12.0 y; 18–81 y	online survey/convenience sampling	self-report measure	4	D		BT	WT	
Wright K.P., 2020 [173]	USA	longitudinal	B: 29 January–4 February 2020; D: 22–29 April 2020	university students	139; F 70.5%	22.2 ± 1.7 y	online survey/convenience sampling	self-report measure	6	D		BT	WT	
Yang G., 2021 [174]	China	cross-sectional	23 February –4 March 2020	general population	2702; F 70.7%	≥18 y, 37.3 ± 12.0 y	online survey/convenience sampling	self-report measure	5	D				
Yang S., 2020 [175]	China	longitudinal	B:23 December 2019–23 January 2020; D: 24–23 February 2020	students	10082; F 71.7%	19.8 ± 2.3 y	online survey/snowball sampling	self-report measure	6	D				
Zalech M., 2021 [176]	Poland	longitudinal	B: 2019; D: 20209	university students	B: 86, F nr; D: 88, F nr	B: 23.13 ± 0.86 y; D: 23.10 ± 1.04	online survey/nr	self-report measure	8	D				
Zheng C., 2020 [177]	China	longitudinal	B: 2019; D: 15–26 April 2020	general population	631 (B/D: 70); F 61.2%	18–35 y; 21.1 ± 2.9 y	online survey/convenience sampling	self-report measure	5	D				
Zhu Q., 2021 [178]	China	cross-sectional	29 March–5 April 2020	general population	889; F 61%	16–70 y; 31.8 ± 11.4 y	online survey/convenience sampling	self-report measure	3	D				

Abbreviations: B: before; D: during; F: female; Y: years; IQR: interquartile range; NR: not reported; D: sleep duration; BT: bedtime; WT: wake-up time; TIB: time in bed; outcomes not included in meta-analysis

**Table 2 ijerph-21-00583-t002:** Percentages of change in sleep duration from before to during the lockdown by country’s area.

Country’s Area	Studies	Percentages of Change in Sleep Duration		
		Change	Decrease	Increase
North America (Canada, USA)	[47,65,85,98,140]	51.4% (95% CI 42.38–60.35; I^2^ = 93.9%; not significant Eggers´s publication bias)	19.8% (95% CI 16.21–23.62; I^2^ = 75.5%; not significant Eggers´s publication bias)	30.5% (95% CI 22.17–39.43; I^2^ = 94.4%: not significant Eggers´s publication bias)
South America (Argentina, Brazil, Mexico, Peru)	[94,109,164,168]	71.8% (95% CI 68.76–74.78; I^2^ = 84.0%; not significant Eggers´s publication bias)	27.6% (95% CI 22.20–33.26; I^2^ = 96.6%; not significant Eggers´s publication bias)	40.3% (95% CI 35.40–45.24; I^2^ = 92.9%; significant Eggers´s publication bias)
Central Asia (Bangladesh, India, Malaysia, Nepal, Pakistan)	[31,79,91,96,99,131,150]	58.0% (95% CI 44.45–71.01; I^2^ = 98.5%; not significant Eggers´s publication bias)	13.4% (95% CI 9.51–17.86; I^2^ = 92.2%; not significant Eggers´s publication bias)	44.1% (95% CI 32.03–56.63; I^2^ = 98.3%; not significant Eggers´s publication bias)
East Asia (China, Japan, Singapore)	[121,172,174,175,178]	45.3% (95% CI 25.13–66.21; I^2^ = 99.9%)	11.2% (95% CI 4.40–20.62; I^2^ = 99.7%)	33.3% (95% CI 19.84–48.33; I^2^ = 99.8%)
West Asia (Iran, Jordan, Kuwait, Lebanon, Saudi Arabia, United Arab Emirates)	[28,29,33,53,60]	55.9% (95% CI 45.40–66.05; I^2^ = 99.1%; not significant Eggers´s publication bias)	18.6% (95% CI 15.43–22.09; I^2^ = 94.6%; not significant Eggers´s publication bias)	36.4% (95% CI 26.10–47.40; I^2^ = 99.2%; not significant Eggers´s publication bias)
Europe (France, Germany, Hungary, Netherlands, Poland, Romania, UK)	[44,46,73,90,97,138,139,142,155,166]	50.0% (95% CI 43.71–56.23; I^2^ = 99.2%; not significant Eggers´s publication bias)	18.2% (95% CI 12.83–24.31; I^2^ = 99.4%; not significant Eggers´s publication bias)	30.8% (95% CI 27.10–34.56; I^2^ = 98.0%; not significant Eggers´s publication bias)
Mediterranean Europe (Catalonia, Cyprus, Greece, Italy, Portugal, Spain, Turkey)	[38,41,54,71,89,101,105,124,154,157]	56.0% (95% CI 41.68–69.78; I^2^ = 99.6%; significant Eggers´s publication bias)	23.6% (95% CI 17.30–30.64; I^2^ = 98.1%; not significant Eggers´s publication bias)	33.8% (95% CI 27.46–40.34; I^2^ = 99.1%; significant Eggers´s publication bias)

**Table 3 ijerph-21-00583-t003:** Synthesis of changes in outcomes from before to during the lockdown.

Author, Year	Outcome	Direction of Change ^a^
	**Duration**	
Abouzid M., 2021 [25]	Significant increase in sleep hours for 53.2% (*p* < 0.001).	↑
Blume C., 2021 [48]	Significant increase in sleep duration by about 13 min (*p* < 0.001).	↑
Casas R., 2022 [55]	Nearly half of the participants reported no change in sleep duration.	≈
Dragun R., 2020 [75]	Significant increase in the median length of sleep duration by 1.5 h (*p* < 0.001).	↑
Falkingham J., 2022 [80]	Increase in the prevalence of sleep loss compared to 2019 (22% vs. 13.9%) particularly marked among the women and the Black, Asian, and the individuals of other minorities (*p* < 0.01).	↓
García-Garro P.A., 2022 [87]	Significant decrease in sleep duration (*p* < 0.001).	↓
Gibson R., 2022 [88]	No significant change in the weighted 24 h sleep duration (*p* = 0.161).	≈
Khojasteh M.R., 2022 [102]	Significant increase in sleep duration (*p* < 0.001).	↑
Kim A.C.H., 2022 [104]	Significant increase in sleep time in the young (18–39 y) and middle-aged participants (40–59 y) (*p* < 0.001). No significant decrease in the older participants (60 ≥ y).	↑Y ↑M ↓O
Kontsevaya A.V., 2021 [106]	Significant decrease in the number of days per week that participants reported not getting enough sleep (from 3.21 ± 2.44 to 2.86 ± 2.57, *p* < 0.001).	↑
Li J.W., 2021 [108]	Significant increase in sleep duration by 0.5 h (*p* < 0.001).	↑
Majumdar P., 2020 [113]	Significant decrease in sleep duration in office workers (*p* < 0.001) and significant increase in sleep duration in students (*p* < 0.001).	↓W ↑S
Pachocka L., 2022 [123]	Decrease in sleep hours.	↓
Pépin J.-L., 2021 [126]	Significant increase in objectively measured total sleep time (*p* < 0.01).	↑
Peterson M., 2021 [128]	Significant increase in sleep duration (*p* = 0.016).	↑
Pouget M., 2022 [132]	No significant change in hours of sleep per night.	≈
Ruiz-Zaldibar C., 2022 [141]	Significant increase in adequate nighttime sleep (7 to 9 h per night) in both the males (*p* = 0.011) and females (*p* < 0.001).	↑
Salfi F., 2021 [144]	Significant difference between the three chronotype groups (evening-type/neither-type/morning type) for the reported sleep duration (*p* < 0.001): the evening-type slept more than the neither-type and morning-type groups.	↑ET
Santos-Miranda E., 2021 [145]	Significant increase in sleep hours (*p* < 0.001).	↑
Souza T.C., 2022 [156]	Significant increase in sleep hours (*p* <0.001).	↑
Tang N.K.Y., 2022 [159]	More participants reported an increase in sleep duration.	↑
Trakada A., 2020 [162]	Significant increase in sleep duration (*p* < 0.001).	↑
Tsigkas G., 2021 [163]	Significant increase in the percentage of people sleeping > 7 h (*p* < 0.001) mainly in the younger persons and in those with a higher income (*p* < 0.001).	↑
Vinogradov O.O., 2022 [169]	No change in sleep duration.	≈
Vollmer C., 2022 [171]	Increase in sleep duration on workdays but not on weekends (*p* < 0.001).	↑
	**Time in bed**	
Czeisler M.É., 2021 [69]	Increased time in bed.	↑
Gibson R., 2022 [88]	Significant increase in time in bed both on workdays and on weekends (*p* < 0.0001).	↑
Pépin J.-L., 2021 [126]	Significant increase in time in bed (*p* < 0.01).	↑
	** *Sleep timing* **	
	**Bedtime**	
Aishworiya R., 2021 [27]	Delay in bedtime.	→
Anastasiou E., 2021 [37]	Delay in bedtime.	→
Azizi A., 2020 [42]	Significant delay in bedtime (*p* < 0.0001).	→
Bottary R., 2022 [50]	Significant delay in bedtime (*p* < 0.001).	→
Elhadi M., 2021 [76]	Significant delay in bedtime (*p* < 0.001).	→
Gibson R., 2022 [88]	Significant delay in bedtime both on workdays and on weekends (*p* < 0.001).	→
Kaizi-Lutu M., 2021 [100]	Among the participants, 36.3% reported an earlier bedtime.	NA
Mónaco E., 2022 [119]	Delay in bedtime.	→
Pépin J.-L., 2021 [126]	No significant delay in bedtime. Greater delay in eveningness compared to morningness chronotypes (*p* < 0.01).	≈
Perez-Carbonell L., 2020 [127]	Among the participants, 30% reported a delay in bedtime.	NA
Sañudo B., 2020 [146]	Earlier bedtime.	←
Souza T.C., 2022 [156]	Significantly earlier bedtime (*p* < 0.0001).	←
Tang N.K.Y., 2022 [159]	Delay in bedtime both in the students and in the young adults.	→
Vinogradov O.O., 2022 [169]	Delay in bedtime.	→
Vollmer C., 2022 [171]	Significant delay in bedtime on workdays but not on weekends, especially in the youngest teachers (*p* < 0.001).	→
	**Wake-up time**	
Aishworiya R., 2021 [27]	Delay in wake-up time both in the mothers and fathers.	→
Anastasiou E., 2021 [37]	Delay in wake-up time.	→
Azizi A., 2020 [42]	Significant delay in wake-up time (*p* < 0.0001).	→
Diz-Ferreira E., 2021 [74]	Significantly earlier wake-up time (*p* < 0.001).	←
Elhadi M., 2021 [76]	Significant delay in wake-up time (*p* < 0.001).	→
Gibson R., 2022 [88]	Significant delay in wake-up time both on workdays (*p* < 0.0001) and on weekends (*p* < 0.001).	→
Kaizi-Lutu M., 2021 [100]	Among the participants, 36.3% reported an earlier wake-up time.	NA
Kontsevaya A.V., 2021 [106]	No significant change in the number of days per week the participants reported an earlier wake-up time.	≈
Mónaco E., 2022 [119]	Delay in wake-up time.	→
Pépin J.-L., 2021 [126]	No significant delay in wake-up time. Greater delay in eveningness compared to morningness chronotypes (*p* < 0.01).	≈
Scarpelli S., 2021 [147]	The majority of the participants (60.9%) reported changes in wake-up time.	NA
Sañudo B., 2020 [146]	Delay in wake-up time.	→
Shahzadi K., 2021 [148]	Delay in wake-up time.	→
Souza T.C., 2022 [156]	Significant delay in wake-up time (*p* < 0.0001).	→
Vinogradov O.O., 2022 [169]	Delay in wake-up time.	→
Vollmer C., 2022 [171]	Significant delay in wake-up time both on workdays (*p* < 0.001) and on weekends (*p* = 0.027).	→
	**Napping habits**	
Alomari M.A., 2021 [33]	Significant decrease in nap hours (*p* < 0.0001).	↓
AMHSI Research Team, 2021 [36]	Significant increase in the percentage of participants taking regular naps (*p* = 0.004).	↑
Azizi A., 2020 [42]	Significant increase in the length of naps (*p* < 0.0001).	↑
Chouchou F., 2021 [63]	Significant increase in the frequency and in the length of naps *(p* < 0.001).	↑
Conte F., 2021 [64]	No significant change in the frequency and in the length of naps.	≈
Felician J., 2022 [81]	Decrease in the percentage of participants taking naps (from 42% to 36%)	↓
Franceschini C., 2020 [84]	Most of the good sleepers did not change or reduce the length of naps while the poor sleepers reported an increase *(p* < 0.001).	↓GS↑PS
García-Esquinas E., 2021 [86]	Decrease in the percentage of participants taking naps (from 65% to 45%).	↓
Gupta R., 2020 [91]	Significant increase in the percentage of participants taking naps (*p* < 0.001).	↑
Kaizi-Lutu M., 2021 [100]	Increase in the frequency of naps.	↑
Leone M.J., 2020 [107]	Significant decrease in the percentage of participants taking naps both on weekdays (from 58.1% to 48.1%, *p* < 0.0001) and on weekends (from 51.3% to 66.3% *p* < 0.0001).	↓
Majumdar P., 2020 [113]	Significant increase in the length of naps both in the students and in the office workers (*p* < 0.05).	↑
Mohsin A., 2021 [118]	Increase in the frequency (from 40.1% to 50.8%) and the length of naps (from 26.6% to 34.9% of nap exceeding one hour).	↑
Mónaco E., 2022 [119]	Increase in the frequency of long naps.	↑
Morin C.M., 2022 [120]	Significant increase in the frequency of naps (almost twice) (*p* < 0.0001).	↑
Petrov M.E., 2021 [129]	Significant increase in the frequency of naps (*p* < 0.001).	↑
Ramirez C., 2022 [136]	Significant increase in the length of naps both on workdays (*p* < 0.001) and on weekends (*p* < 0.01).	↑
Salfi F., 2021 [144]	The majority of the participants maintained unchanged napping habits (64.7%). A significantly higher percentage of the evening-type subjects reported changes in napping habits compared to the morning-type and neither-type chronotypes (*p* < 0.01).	≈
Santos-Miranda E., 2021 [145]	Increase in the frequency of naps (*p* = 0.051). Significant increase in the length of naps (*p* = 0.034).	↑
Scarpelli S., 2021 [147]	The majority of the participants (60.8%) reported changes in napping habits.	NA
Sinha M., 2020a [152]	The majority of the participants reported an increase in the frequency of naps.	↑
Valiensi S.M., 2022 [165]	The majority of the participants reported no change in the frequency of naps and a decrease in the length of naps	≈F↓L
Vinogradov O.O., 2022 [169]	Slight increase in the percentage of the participants taking naps (from 62.8 to 69.8%); no change in the length of naps.	↑P≈L

Abbreviations: ET: evening-type chronotype; F, frequency; GS: good sleepers; L, length; M: middle-aged; NA, not applicable; O: older; P: percentage; PS: poor sleepers; W, workers; S, students; Y: young. ^a^ ↑ indicates an increase, ↓ indicates a decrease, ≈ indicates no change, → indicates a delay in a sleep timing, ← indicates earlier sleep timing.

## Data Availability

No new data were created in this study. Data sharing is not applicable to this article.

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
