# Peer review of "Changes in Sleep Duration and Sleep Timing in the General Population from before to during the First COVID-19 Lockdown: A Systematic Review and Meta-Analysis"

_ijerph, 2024, doi:10.3390/ijerph21050583_

Round 1
Reviewer 1 Report
Comments and Suggestions for Authors
Dear authors,
The manuscript provides a detailed review of studies that have examined sleep changes during the first COVID-19-related lockdown. To my knowledge, the review and the meta-analytical methods do not raise any concerns. The only concern regards the choice to investigate only the studies on the first lockdown, whereas a comparative investigation of the different COVID-19 phases would have been more interesting and constructive. On the other hand, the authors investigated napping habits during COVID-19, which have been poorly investigated by other reviews, and therefore present a novelty compared to the current literature.
The manuscript is limited by a relatively poor novelty, as well as by the choice to only consider studies conducted during the first lockdown, that have already been largely reviewed. However, it also relies on strength points, such as the inclusion of changes in napping habits, which have been overlooked by other reviews, and the analytical comparison of studies with high and low risk of bias.
I just have one concern about the decision to only consider studies related to the first lockdown. This decision should be better explained.
The review offers a clear overview of the studies presented that could help the development of screening programs and evidence-based interventions for sleep disturbances in times of crisis. Furthermore, the findings presented highlight the important role of sleep duration, sleep schedules, and napping habits in the assessment of sleep quality.
Since one of the most innovative aspects of this review is the assessment of changes in napping habits, the introduction could provide more information on this topic and its significance.
In interpreting the results, I would consider more in-depth the possible role of other factors, such as age, gender, and the region of residence, which have been proven to modulate the effects of COVID-19 on sleep (e.g., Cerasuolo et al., 2021; Cellini et al., 2021; Jahrami et al., 2022).
I may suggest a brief paragraph comparing sleep changes during the first lockdown with the other phases of the COVID-19 pandemic, to detect a possible pattern of sleep changes from before to during the different lockdown phases.
Although limited to the first COVID-19 lockdown, this manuscript enhances the existing literature by providing a detailed investigation of changes in napping habits from before to during the COVID-19 pandemic.
The role of napping for health is still a debated topic. Very interestingly, the manuscript investigated the changes in napping habits from before to during the first COVID-19 lockdown. Offering a broader scope to this topic, and discussing its health implications in more detail, could increase the novelty of the manuscript.
Given the well-studied topic (i.e., sleep in COVID-19), the quality of the manuscript might be improved by placing more emphasis on the innovative topics neglected by other reviews.
The authors could enhance the manuscript by 1) highlighting the more innovative aspects (like the changes in napping habits); 2) conducting a more in-depth analysis of the studies included in the review (e.g., by investigating the possible differences by age and/or gender and/or region); 3) comparing their findings with the other phases of COVID-19 pandemic; 4) discussing the possible role of the drastic sleep changes during the first COVID-19 lockdown for the development of the long COVID-19 syndrome.
Additional minor comments to address.
1. In the abstract the authors should better clarify the direction of the results (e.g., "Sleep duration and TIB were slightly increased" during the lockdown). Also, when possible provide the amount of these modifications.
2. Row 148: the point of the list is missing.
3. The three synthesis paragraphs at the end of each results section should be aggregated into a single paragraph to be placed before the discussion.
4. Strengths and limitations: this section should only include the review's strengths and limitations, not those related to the studies (e.g., the use of self-report measures), which should be addressed at the beginning of the discussion.
5. Conclusion and throughout: since most of the studies examined were cross-sectional, terms such as "association" or "relationship" should be preferred to "impact".
Reviewer 2 Report
Comments and Suggestions for Authors
General concept comments
This systematic review looks at several parameters regarding changes in sleep behavior before and after the COVID-19 lockdown, namely sleep duration, time in bed, sleep timing, as well as the proportion of long and short sleep and napping occurrence. The focus were studies in the general adult population, and there was a broad representation of geographical areas. The main conclusions agreed with what has been reported previously, which is an increase in sleep duration and delayed sleep timing. Furthermore, the additional parameters analyzed such as occurrence of atypical sleep duration are a strength of this report. The methods used are appropriate, and their description go according to guidelines and are sufficient and clear.
I find portions of the discussion of the review to be unrelated to the topic at hand, especially the section about the consequences of quantitative sleep parameter alterations on general health. I would suggest narrowing down this section in particular, since it falls out of reach of the conclusions of the review.
Specific comments
Line 46-47. “Changes in routines disrupted circadian rhythms and energy balance, affecting various biological clock regulators”. I would use the term “daily rhythms” instead of “circadian rhythms” since the concept of circadian rhythm implies endogenous mechanisms which these studies usually do not measure, as they are based on self-reports.
The reference here is to a review article (Caroppo et al, Changes in Lifestyle during COVID-19 Pandemic and Consequences on Mental Health) which in turn only references an article regarding external synchronization of circadian clocks (Baquerizo-Sedano et al, Anti-COVID-19 measures threaten our healthy body weight: Changes in sleep and external synchronizers of circadian clocks during confinement). I recommend referencing this paper instead, which indeed found a reduction in light exposure, physical activity and eating schedules during lockdown, which are synchronizers.
Line 48. “As the circadian system regulates the sleep-wake cycle…” I would add “the circadian system partially regulates…” since the consensus in the field is that there is an important homeostatic component to sleep regulation.
Reviewer 3 Report
Comments and Suggestions for Authors
Title: Changes in sleep duration and sleep timing in the general pop-2 ulation from before to during the first COVID-19 lockdown: A 3 systematic review and meta-analysis
The present study provides a systematic review and meta-analysis of quantitative sleep parameters during the first lockdown from March 2020 with reference to the pre-lockdown period. The authors have examined sleep duration, time in bed (TIB), and sleep timing (bedtime and wake-up time), napping, along with percentages of atypical sleep duration before and during the lockdown, change in sleep duration and sleep timing. Their review found that sleep duration and TIB slightly increased. The manuscript is very lucidly written with comprehensive statistical analysis. It is an exhaustive literature search and review which increases faith in the results obtained by the meta-analysis. In my opinion, the manuscript should be accepted for publication after minor revision since it contributes significantly to the existing literature on sleep during the COVID-19 pandemic. There are some comments/ concerns that need to be addressed.
1. Page 2, Line 46: Changes in routine would disrupt sleep patterns and other behavioral patterns but not disrupt the circadian rhythm. Please refer to the cited article again. Circadian disruption is altogether different and has much more drastic effects on the temporal organization of biological functions.
2. Page 3, l09: “….the quantity of time….” Please change to amount of time.
3. Page 3, L 111: Please correct the definition of bedtime. Instead of the time the person falls asleep, the time the person goes to bed is more appropriate since bedtime and sleep time may vary.
4. The authors can use the term daytime sleepiness to indicate napping.
5. Page 3, l37: Typos. “Instead” is replaced with additionally or another appropriate word/ phrase.
6. Page 39, L453-455: The authors argue that the increase in sleep duration does not necessarily reflect an increase in sleep quality. Their previous review concluded worsening sleep quality during the first COVID lockdown. The authors also need to discuss studies that indicate that increased sleep duration would enhance sleep quality. It can be argued that during lockdown, many people may have got more opportunities to relax, particularly in urban areas with hectic work schedules. I would suggest the authors do a meta-analysis by segregating the data based on place of residence (urban vs. rural cohort) or based on age and sex, or based on income groups since the stress levels experienced by different cohorts during lockdown would be different.
7. Page 39. Section 4.1: The discussion is heavily weighed on the negative effects of changing sleep characteristics. The authors are suggested to provide positive impacts of varying sleep parameters on the immune system and general health.
8. Page 41: Conclusion section: The authors are stretching the impact on sleep characteristics, viz. duration of sleep, TIB, and daytime napping to sleep-wake cycle disruption. Longer sleep duration and time in bed do not necessarily reflect disruption in the sleep-wake cycle. I would suggest the authors be cautious in interpreting the data since they have not analyzed sleep cycle disruption in this study.
Reviewer 4 Report
Comments and Suggestions for Authors
The PRISMA flow chart should be revised and corrected. For example, after subtracting 965 articles from 6289, the result is 5324! Anyway, the other boxes should be checked too.
